# Duality between predictability and reconstructability in complex systems

Charles Murphy [1,2] ✉, Vincent Thibeault [1,2], Antoine Allard [1,2] & Patrick Desrosiers [1,2,3] ✉

Predicting the evolution of a large system of units using its structure of interaction is a fundamental problem in complex system theory. And so is the problem of reconstructing the structure of interaction from temporal observations. Here, we find an intricate relationship between predictability and reconstructability using an information-theoretical point of view. We use the mutual information between a random graph and a stochastic process evolving on this random graph to quantify their codependence. Then, we show how the uncertainty coefficients, which are intimately related to that mutual information, quantify our ability to reconstruct a graph from an observed time series, and our ability to predict the evolution of a process from the structure of its interactions. We provide analytical calculations of the uncertainty coefficients for many different systems, including continuous deterministic systems, and describe a numerical procedure when exact calculations are intractable. Interestingly, we find that predictability and reconstructability, even though closely connected by the mutual information, can behave differently, even in a dual manner. We prove how such duality universally emerges when changing the number of steps in the process. Finally, we provide evidence that predictability-reconstruction dualities may exist in dynamical processes on real networks close to criticality.

The relationship between structure and function is fundamental in complex systems[1–3], and important efforts have been invested in developing network models to better understand it. In particular, models of dynamics on networks[4–7] have been proposed to assess the influence of network structure over the temporal evolution of the activity in the system. In turn, data-driven models[8, 9], dimension-reduction techniques[10–14], and mean-field frameworks[15–19] have deepened our predictive capabilities. Among other things, these theoretical approaches have shed light on the relationship between dynamics criticality and many network properties such as the degree distribution[15,17], the eigenvalue spectrum[20–22], and their community structure[23,24]. Fundamentally, these contributions justify our

inclination for measuring and using real-world networks as a proxy to predict the behavior of complex systems.

Models of dynamics on networks have also been used as reverse engineering tools for network reconstruction[25], when the networks of interactions are unavailable, noisy[26–28] or faulty[29]. The network reconstruction problem has stimulated many technical contributions[30]: Thresholding matrices built from correlation[31] or other more sophisticated measures[32,33] of time series, Bayesian inference of graphical models[34–39], and models of dynamics on networks[40], among others. These techniques are widely used (e.g., in neuroscience[41–43], genetics[44], epidemiology[40,45], and finance[46]) to reconstruct interaction networks on which network science tools can then be applied.

[1]Département de physique, de génie physique et d'optique, Université Laval, Québec, QC G1V 0A6, Canada. [2]Centre interdisciplinaire en modélisation mathématique, Université Laval, Québec, QC G1V 0A6, Canada. [3]Centre de recherche CERVO, Québec, QC G1J 2G3, Canada. ✉e-mail: charles.murphy.1@ulaval.ca; patrick.desrosiers@phy.ulaval.ca

Interestingly, dynamics prediction and network reconstruction are usually considered separately, even though they are related to one another. The emergent field of network neuroscience[47,48] is perhaps the most actively using both notions: Network reconstruction for building brain connectomes from functional time series, then dynamics prediction for inferring various brain disorders from these connectomes[49,50]. Recent theoretical works have also taken advantage of these notions to suggest that dynamics may hardly depend on the structure. In ref. 51, it was shown that time series generated by a deterministic dynamics evolving on a specific graph can be accurately predicted by a broad range of other graphs. These findings highlight how poor our intuition can be with regard to the relationship between predictability and reconstructability. Furthermore, recent breakthroughs in deep learning on graphs have benefited from proxy network substrates to enhance the predictive power of their models[52,53], with applications in epidemiology[9], and pharmaceutics[54,55]. However, the use of graph neural networks and those proxy network substrates is only supported by numerical evidence and lacks a rigorous theoretical justification. As a result, their enhanced predictability remains to be fully corroborated. There is therefore a need for a solid, theoretical foundation of reconstructability, predictability, and their relationship in networked systems.

In this work, we establish a rigorous framework that lays such a foundation based on information theory. Information theory has been regularly applied to networks and dynamics in the past. In network science, it has been used to characterize random graph ensembles[56–58]—e.g. the configuration model[59,60] and stochastic block models[61,62]—, to develop network null models[63] and to perform community detection[64,65]. In stochastic dynamical systems, information-theoretical measures have been proposed to quantify their

predictability[66–73], complexity[74,75] and causal emergence[76]. In statistical mechanics, information transmission has been shown to reach a maximum value near the critical point of spin systems in equilibrium[77,78].

Our objective is to combine these ideas into a single framework, motivated by recent works involving spin dynamics on lattices[79,80] and deterministic dynamics[51]. Our contributions are fourfold. First, we use mutual information between structure and dynamics as a foundation for our general framework to quantify the structure-function relationship in complex systems. Second, this codependence naturally leads to the definition of measures of predictability and reconstructability. Doing so allows us to conceptually unify prediction and reconstruction problems, i.e., two classes of problems that are usually treated separately. Third, we design efficient numerical techniques for evaluating these measures on large systems. Finally, we identify a new phenomenon—a duality—where our prediction and reconstruction capabilities can vary in opposite directions. These findings further our understanding of the complexity of modeling networked complex systems, such as the brain, where both prediction and reconstruction techniques play critical roles.

## Results
### Information theory of dynamics on random graphs
Let us consider a random graph $G$ whose support, $\mathcal{G}$, consists in the set of all graphs of $N$ vertices, each of which has its respective nonzero probability $P(G = g)$ with $g \in \mathcal{G}$. In our framework, $P(G)$ can be any graph distribution and reflects, from a Bayesian perspective, our prior knowledge of the structure of the system. We also consider a general discrete-time stochastic process (also called dynamics hereafter) with $T$ time steps evolving on a realization of $G$ and representing the

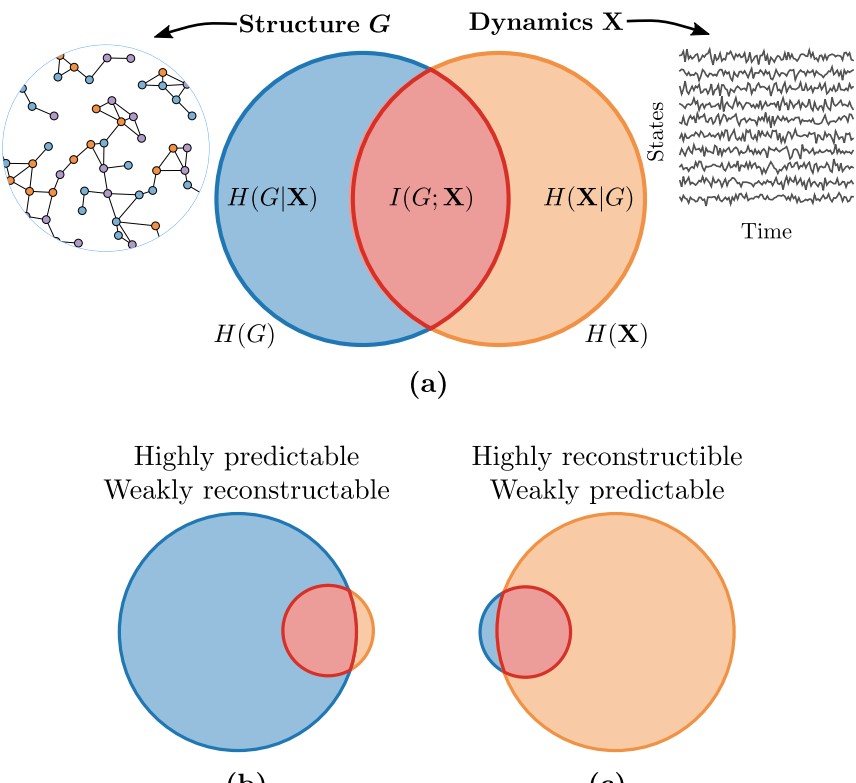

**Fig. 1 | Information diagram of dynamics on random graphs. a** Areas represent amounts of information: The entropies related to $G$ are shown on the left in blue and those related to **X** are on the right in orange. Mutual information---the red intersection of **X** and $G$---corresponds to the information shared by both $G$ and **X**. **b** The highly predictable / weakly reconstructable scenario, where $H(G) \gg H(\mathbf{X})$

meaning that $I(\mathbf{X}; G)$ contains most of the information related to the dynamics, but only a small fraction of the information related to the graph. **c** The reverse scenario, i.e., highly reconstructable / weakly predictable, where $H(\mathbf{X}) \gg H(G)$ meaning that $I(\mathbf{X}; G)$ contains most of the information related to the graph, but only a small fraction of the information related to the dynamics.

possible states of the system. More precisely, we denote $P(\mathbf{X}|G)$ the probability of a random and discrete-state time series $\mathbf{X} = (X_{i,t})_{i,t}$ conditioned on $G$, where $X_{i,t}$ is the random state, with discrete support $\Omega$, of vertex $i \in \{1,...,N\}$ at time $t \in \{1,...,T\}$. We stress that $\mathbf{X}$ is at this point any stochastic process be it Markovian or not. The initial condition of the process is $\mathbf{X}_1 = (X_{i,1})_i$. While we only exposed our framework in terms of discrete-time and discrete-state processes, it can be used for continuous-state deterministic dynamics (see Supplementary Note III) and in principle, it can also be generalized to continuous-state stochastic processes by considering a probability density function $\rho(\mathbf{X}|G)$.

The variables $\mathbf{X}$ and $G$ form themselves a Bayesian network $G \to \mathbf{X}$, where the arrow indicates conditional dependence[81]. From this model, we are interested in the mutual information between $\mathbf{X}$ and $G$−denoted $I(\mathbf{X};G)$−which is a symmetric measure that quantifies the codependence between the dynamics $\mathbf{X}$ and the structure $G$[82], where $I(\mathbf{X};G) = 0$ when they are independent. It is equivalently given by

$$I(\mathbf{X};G) = H(\mathbf{X}) - H(\mathbf{X}|G) \tag{1a}$$

$$= H(G) - H(G|\mathbf{X}), \tag{1b}$$

where $H(G) = -\langle \log P(G) \rangle$ and $H(\mathbf{X}) = -\langle \log P(\mathbf{X}) \rangle$ are respectively the marginal entropies of $G$ and $\mathbf{X}$, and $H(G|\mathbf{X}) = -\langle \log P(G|\mathbf{X}) \rangle$ and $H(\mathbf{X}|G) = -\langle \log P(\mathbf{X}|G) \rangle$ are their corresponding conditional entropies. In the previous equations, the marginal distribution for $\mathbf{X}$, the evidence, is defined as $P(\mathbf{X}) = \sum_{g \in \mathcal{G}} P(G = g) P(\mathbf{X}|G = g)$, and the posterior is obtained from Bayes' theorem as $P(G|\mathbf{X}) = P(G) P(\mathbf{X}|G)/P(\mathbf{X})$, using the given graph prior $P(G)$ and the dynamics likelihood $P(\mathbf{X}|G)$. $I(\mathbf{X};G)$ is a non-negative measure bounded by $0 \le I(\mathbf{X};G) \le \min\{H(G), H(\mathbf{X})\}$. Figure 1a provides an illustration of Eq. (1) in terms of information diagrams.

The measures presented in Eq. (1) and above can all be interpreted in the context of information theory. Information is generally measured in bits which in turn is interpreted as a minimal number of binary −i.e., yes/no−questions needed to convey it. While entropy measures the uncertainty of random variables like $\mathbf{X}$ and $G$, i.e., the minimal number of bits of information needed to determine their value, mutual information represents the reduction in uncertainty about one variable when the other is known. The fact that it is symmetric means that this reduction goes both ways: The reduction in the dynamics uncertainty when the structure is known is equal to that of the structure when the dynamics is known. Hence, mutual information measures the amount of information shared by both $\mathbf{X}$ and $G$.

As an illustration, let us consider the physical example of a spin system that depends on $G$ through a coupling parameter $J \ge 0$, where the spins are more (large $J$) or less (small $J$) likely to align with their first neighbors in $G$. At $J = 0$, the spins are completely uncorrelated and flip with probability $\frac{1}{2}$. In this case, $H(\mathbf{X}|G) = NT$ bits, corresponding to the maximum entropy of $\mathbf{X}$: We need precisely one binary question for each spin at each time for a given structure $G$−e.g., is the spin of vertex $i$ at time $t$ up? When $J > 0$, correlation is introduced between connected spins. As a result, a single question about the spin of vertex $i$ at time $t$ can provide additional information about the spins of other vertices at other times, and thus, $H(\mathbf{X}|G) < NT$. The interpretation of $H(\mathbf{X})$ is analogous to that of $H(\mathbf{X}|G)$, as it measures the number of binary questions needed to determine $\mathbf{X}$ when the graph is unknown. From this perspective, the mutual information $I(\mathbf{X};G)$, as expressed by the difference between $H(\mathbf{X})$ and $H(\mathbf{X}|G)$, is the reduction in the number of questions needed to predict $\mathbf{X}$ ensuing from the knowledge of $G$. Hence, $I(\mathbf{X};G)$ measures to which extent the knowledge of the graph $G$ improves our ability to forecast $\mathbf{X}$, i.e. its temporal predictability.

Similar observations can be made from the structural perspective. Suppose that $\mathbf{X}$ is the spin dynamics mentioned previously and $G$ is a random graph, where each edge exists independently with probability

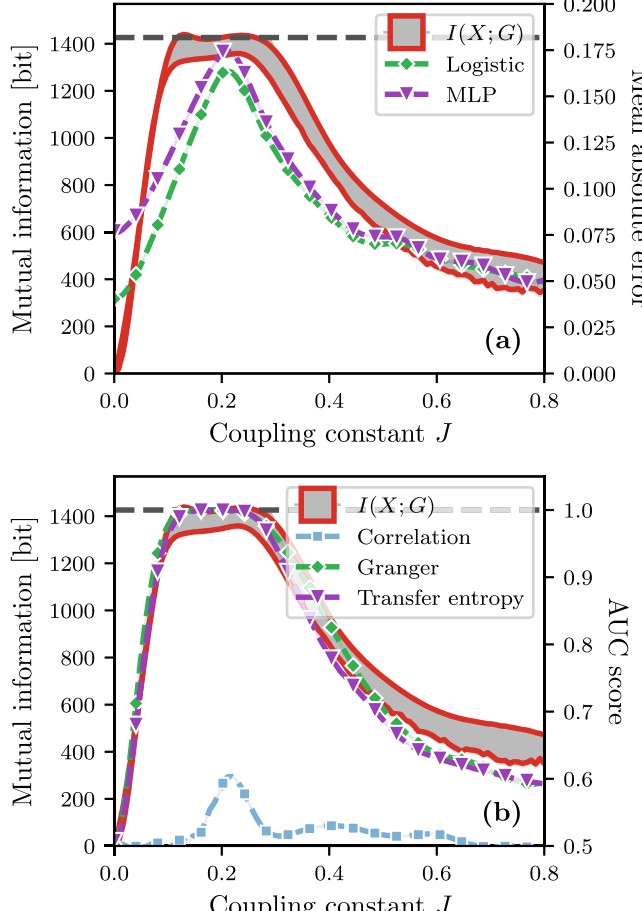

**Fig. 2 | Comparison between the mutual information and algorithm performance measures. a** Prediction algorithms and **b** reconstruction algorithms. This comparison is performed with time series of length $T = 100$ generated with the Glauber dynamics evolving on Erdős-Rényi graphs with $N = 100$ nodes and $M = 250$ edges, for different coupling constants $J$. **a** The mean absolute error between the true transition probabilities used in $P(\mathbf{X}|G)$ and the ones predicted by different graph-independent models models: a logistic regression (green diamonds) and a multilayer perceptron (MLP, purple triangles). **b** The average area under the curve (AUC) of the receiver operating characteristic (ROC) curve for different reconstruction algorithms: the correlation matrix method[31] (light blue squares), the Granger causality method[32] (green diamonds), and the transfer entropy method[33] (purple triangles). In both panels, we use two axes to represent (left axis) $I(\mathbf{X};G)$, denoted by the gray area bounded by the two biased estimators (red lines, see "Estimators of the mutual information" section), and (right axis) the performance measures; the maximum of $I(\mathbf{X};G)$ is shown with the horizontal dashed line. See "Performance of prediction and reconstruction algorithms" section for further details.

$p$. This yields $H(G) = -\binom{N}{2}[p \log p + (1-p) \log(1-p)]$, where $\binom{N}{2}$ is the total number of possible undirected edges. When $p = \frac{1}{2}$, we have $H(G) = \binom{N}{2}$ bits, which is again the maximum entropy of $G$. We therefore need precisely one binary question for each of the $\binom{N}{2}$ edges in the graph−e.g., is there an edge between $i$ and $j$?−to completely determine its state. When the dynamics $\mathbf{X}$ is known, $H(G|\mathbf{X})$ is interpreted similarly to $H(G)$, but also takes into account the observation of the spins $\mathbf{X}$ which introduces correlation between the edges of $G$. As a result, each bit can provide information about more than one edge, even in the case $p = \frac{1}{2}$ where we a priori need one bit per possible

edge to fully reconstruct $G$. Consequently, the knowledge of $\mathbf{X}$ reduces uncertainty about $G$ (i.e., $H(G|\mathbf{X}) \leq H(G)$, see ref. 82, Theorem 2.6.5), and therefore allows for its reconstruction; $I(\mathbf{X}; G)$ thus measures the reconstructability of $G$, i.e. the extent to which information about $G$ can be revealed from $\mathbf{X}$.

In practice, $I(\mathbf{X}; G)$ can be used to explain the performance of both prediction and reconstruction algorithms (see "Performance of prediction and reconstruction algorithms" section for further detail). From the prediction perspective, it quantifies the sensitivity of the time series to the structure of interactions prescribed by $G$, i.e., the gain in predictability of including $G$ for the extrapolation of $\mathbf{X}$. This can be measured by comparing the true transition probabilities of the process $\mathbf{X}$ as given by the conditional model $P(\mathbf{X}|G)$, with those predicted by models that do not include $G$ in their predictions. This experiment was performed in ref. 51 for deterministic dynamics on graphs, to show that high prediction accuracy of time series can sometimes be achieved without the knowledge of the true graph. In Fig. 2a, we use the mean absolute error—the same measure as in ref. 51—to perform the comparison. In turn, we associate the high predictive capabilities of the true conditional model where the error with the graph-independent model is high. Likewise, $I(\mathbf{X}; G)$ provides strong insights into the reconstruction accuracy of algorithms such as the transfer entropy method[33] (Fig. 2b). By interpreting the reconstruction problem as a binary classification, we are allowed to quantify the reconstruction accuracy with the area under the curve (AUC) of the receiver operating characteristic (ROC) curve. In all cases, $I(\mathbf{X}; G)$ peaks in the same coupling interval as the different reconstruction methods even if the two measures are a priori different.

The mutual information $I(\mathbf{X}; G)$ is, therefore, both a measure of predictability and reconstructability, thereby unifying these two concepts. We say that a system is perfectly predictable when the mutual information contains all the information about $\mathbf{X}$, that is when $I(\mathbf{X}; G) = H(\mathbf{X})$ (see Fig. 1b). Likewise, we say that it is perfectly reconstructable when $I(\mathbf{X}; G) = H(G)$ (see Fig. 1c). Consequently, whenever $I(\mathbf{X}; G) > 0$, we expect the system to be predictable and reconstructable to a certain degree. Otherwise, when $I(\mathbf{X}; G) = 0$, the system is said both unpredictable and unreconstructable. Yet, $I(\mathbf{X}; G)$ by itself is hardly comparable from one system to another. Indeed, a specific value of $I(\mathbf{X}; G)$ may correspond to opposing scenarios when it comes to predictability and reconstructability, as shown in Fig. 1b, c. Thus, it is more convenient to use normalized quantities such as the uncertainty coefficients

$$U(\mathbf{X}\,|\,G) = \frac{I(\mathbf{X}; G)}{H(\mathbf{X})}, \tag{2a}$$

$$U(G\,|\,\mathbf{X}) = \frac{I(\mathbf{X}; G)}{H(G)}, \tag{2b}$$

which are bounded between 0 and 1. Contrary to $I(\mathbf{X}; G)$, $U(\mathbf{X}\,|\,G)$ and $U(G\,|\,\mathbf{X})$ represent relative amount of information. For instance, $U(G\,|\,\mathbf{X}) = 1$ implies that $I(\mathbf{X}; G) = H(G)$, which in principle means that perfect reconstruction can be achieved as all the information of $G$ is contained in $\mathbf{X}$. Likewise, $U(\mathbf{X}\,|\,G) = 1$ means that $I(\mathbf{X}; G) = H(\mathbf{X})$, which indicates that all the information in $\mathbf{X}$ is determined by $G$: a perfectly accurate prediction of $\mathbf{X}$ can be made with $G$ alone. This maximum value is guaranteed when $\mathbf{X}$ is deterministic and there is only one initial condition (see Supplementary Note III). Having $I(\mathbf{X}; G) = 0$ implies that $U(\mathbf{X}\,|\,G) = U(G\,|\,\mathbf{X}) = 0$, which again means that $G$ and $\mathbf{X}$ are independent. Any value in-between of $U(\mathbf{X}\,|\,G)$ and $U(G\,|\,\mathbf{X})$ represents different degrees of predictability and reconstructability, respectively.

The "Simple example" section will present simple concrete examples to provide a better intuition about these concepts. Before we get to these examples, we investigate the influence of the knowledge of

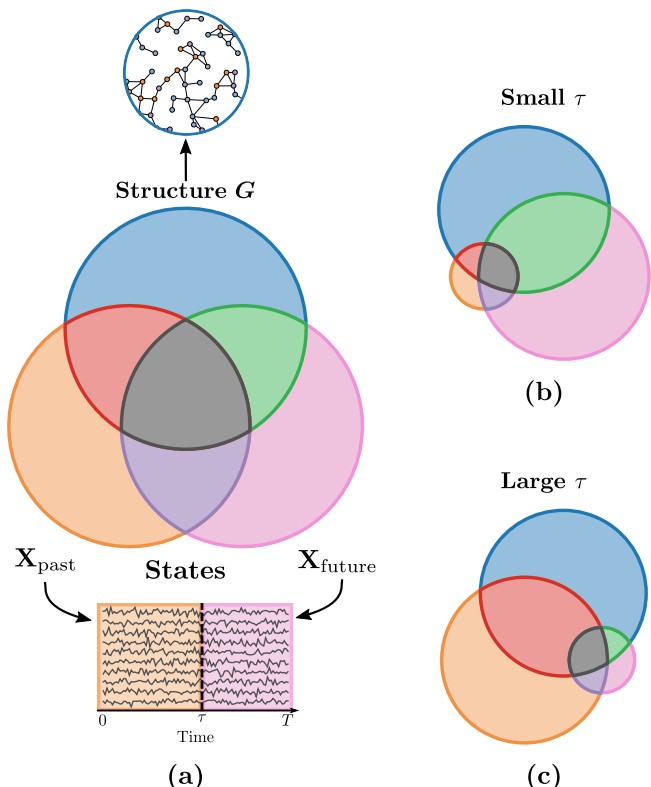

**Fig. 3 | Information diagrams for the past-dependent information measures.** In **a**, we show the information diagram of the random variable triplet $(\mathbf{X}_{\text{past}}, \mathbf{X}_{\text{future}}, G)$, where $\mathbf{X}_{\text{past}}$ represents the past states, $\mathbf{X}_{\text{future}}$, the future states, and $G$, the structure of the system. The quantities of interest are $I(\mathbf{X}_{\text{future}}; G|\mathbf{X}_{\text{past}})$ indicated by the green set, $H(\mathbf{X}_{\text{future}}|\mathbf{X}_{\text{past}})$ shown by the union of the pink and green sets, and $H(G|\mathbf{X}_{\text{past}})$, represented by the union of the blue and green sets. **b, c** Two extreme scenarios where the length of the past $\tau$ is small and large, which illustrates how the different information measures change with $\tau$.

the past of $\mathbf{X}$ over the relationship between its future and its structure, as measured through reconstructability and predictability.

## Past-dependent mutual information

It is often the case that predictability measures the sensitivity to the initial conditions of a process $\mathbf{X}$. For instance, refs. 66,68,83,84 used different versions of the mutual information between $X_1$ and $\mathbf{X}$ as a direct measure of predictability. Then, a system is more predictable if the past allows it to better predict the future. In this spirit, we generalize our framework in such a way that the mutual information between the process $\mathbf{X}$ and its structure $G$ includes some information about the past of $\mathbf{X}$.

We define $\mathbf{X}_{\text{past}}$ as the past of $\mathbf{X}$ and $\mathbf{X}_{\text{future}}$ as its future, such that $\mathbf{X} = (\mathbf{X}_{\text{past}}, \mathbf{X}_{\text{future}})$, see Fig. 3a. We define $\tau$ as the length of $\mathbf{X}_{\text{past}}$ and $T - \tau$ as the length of the future $\mathbf{X}_{\text{future}}$. Our measure of interest in this case is $I(\mathbf{X}_{\text{future}}; G|\mathbf{X}_{\text{past}})$, and it is equal to

$$I(\mathbf{X}_{\text{future}}; G|\mathbf{X}_{\text{past}}) = I(\mathbf{X}; G) - I(\mathbf{X}_{\text{past}}; G), \tag{3}$$

which is a conditional mutual information—the green intersection in Fig. 3a. In turn, a small $\tau$ includes less contribution to the observed past, which leads to a scenario increasingly similar to that presented in "Information theory of dynamics on random graphs" section as shown by Fig. 3b. As $\tau$ gets larger, more contribution is left to $\mathbf{X}_{\text{past}}$ resulting in a smaller $I(\mathbf{X}_{\text{future}}; G|\mathbf{X}_{\text{past}})$, even though the total mutual information $I(\mathbf{X}; G)$—the union of the red, green, and gray sets—is large (see Fig. 3c). Similarly to "Information theory of dynamics on random graphs" section, we then define the partial uncertainty coefficients, bounded

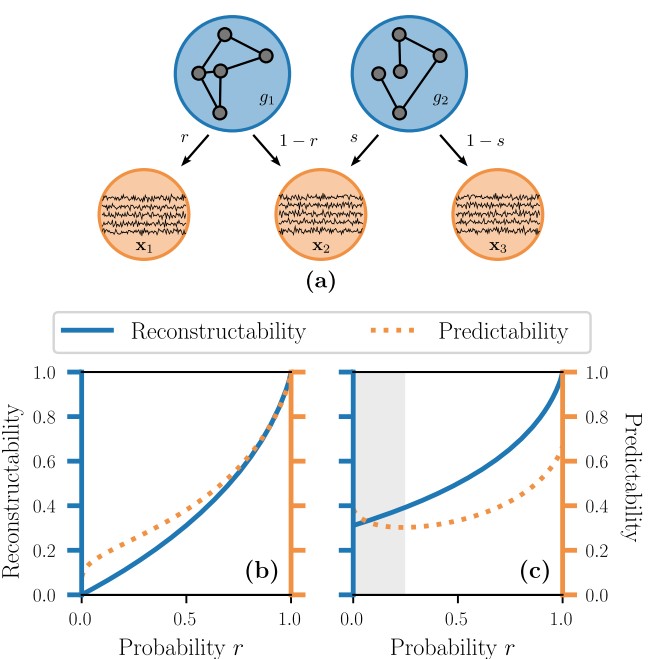

(a)

**Fig. 4 | Example with two graphs and three time series. a** The stochastic system consists of two possible graphs, $g_1$ and $g_2$, and three possible time series, $\mathbf{x_1}$, $\mathbf{x_2}$, and $\mathbf{x_3}$. **b, c** The reconstructability $U(G|\mathbf{X})$ (solid blue line) and the predictability $U(\mathbf{X}|G)$ (dashed orange line) are shown for $s=1$ (**b**) and $s=\frac{1}{2}$ (**c**). The shaded area in (**c**) indicates the region where $U(\mathbf{X}|G)$ and $\mathbf{U}(G|\mathbf{X})$ vary in opposite directions.

between 0 and 1:

$$U(\mathbf{X}_{\text{future}}|G;\mathbf{X}_{\text{past}}) = \frac{I(\mathbf{X}_{\text{future}};G|\mathbf{X}_{\text{past}})}{H(\mathbf{X}_{\text{future}}|\mathbf{X}_{\text{past}})}, \quad (4a)$$

$$U(G|\mathbf{X}_{\text{future}};\mathbf{X}_{\text{past}}) = \frac{I(\mathbf{X}_{\text{future}};G|\mathbf{X}_{\text{past}})}{H(G|\mathbf{X}_{\text{past}})}, \quad (4b)$$

measuring the partial predictability of $\mathbf{X}_{\text{future}}$ from $G$ and partial reconstructability of $G$ given $\mathbf{X}_{\text{future}}$, respectively. The above quantities can be expressed in terms of previously visited ones. For instance, in Eq. (3), $I(\mathbf{X};G)$ and $I(\mathbf{X}_{\text{past}};G)$ can be expressed using Eq. (1). Likewise, the normalizing factor $H(\mathbf{X}_{\text{future}}|\mathbf{X}_{\text{past}})$ is expressed in terms of state entropies using the joint entropy $H(\mathbf{X}) = H(\mathbf{X}_{\text{past}},\mathbf{X}_{\text{future}})$, i.e., $H(\mathbf{X}_{\text{future}}|\mathbf{X}_{\text{past}}) = H(\mathbf{X}) - H(\mathbf{X}_{\text{past}})$. And finally, $H(G|\mathbf{X}_{\text{past}})$ is evaluated similarly to $H(G|\mathbf{X})$.

Whereas the interpretation of the partial uncertainty coefficients is analogous to those presented in the previous section, they nevertheless measure conceptually different quantities. Indeed, by using $I(\mathbf{X}_{\text{future}};G|\mathbf{X}_{\text{past}})$, it is implied that the information about the past has been removed from the total mutual information between $\mathbf{X}$ and $G$. As a result, the partial predictability $U(\mathbf{X}_{\text{future}}|G;\mathbf{X}_{\text{past}})$ measures the gain in predictability over $\mathbf{X}_{\text{future}}$ when including $G$ in the prediction, compared to a model which only uses $\mathbf{X}_{\text{past}}$. Additionally, the removed information likely includes some information about $G$, since $I(\mathbf{X}_{\text{past}};G) \geq 0$. Hence, the partial reconstructability, as defined by $U(G|\mathbf{X}_{\text{future}};\mathbf{X}_{\text{past}})$, measures the reconstructability of the remaining information about $G$ when observing $\mathbf{X}_{\text{future}}$, i.e., information which has not been unveiled from the observation of $\mathbf{X}_{\text{past}}$.

In essence, for some $\xi > 0$, the case $\tau = 1$ with $T = \xi + 1$ is similar to the case $\tau = T - \xi$ with $T > \xi$ since $\mathbf{X}_{\text{future}}$ have the same length $\xi$ in both cases. From a reconstruction perspective, they quantify the reconstructability of $G$ from a process with $\xi$ time steps. However, the reconstructed information is quite different in both cases, since with

$\tau = 1$ and $T = \xi + 1$ no prior information is given—assuming that the initial conditions $\mathbf{X}_1$ are independent from $G$—, while a lot of information has already been processed when $\tau = T - \xi$. Furthermore, increasing $\tau$ draws our attention away from the actual relationship between $\mathbf{X}$ and $G$ of interest, since this relationship should exclude all information about $\mathbf{X}_{\text{past}}$. For this reason, we will mostly focus on the case $\tau = 1$ in the remainder of the paper.

## Simple example

The interpretation of reconstructability and predictability in terms of $U(G|\mathbf{X})$ and $U(\mathbf{X}|G)$ can be grasped more firmly through an elementary example. We consider a system where only two graphs are possible, namely $g_1$ and $g_2$, such that $P(g_1) = p$ and $P(g_2) = 1 - p$ (see Fig. 4a). The entropy of this graph is therefore $H(G) = \mathcal{H}(p)$, where $\mathcal{H}(p) = -p\log p - (1-p)\log(1-p)$ is the binary entropy. These two graphs can generate together three outcomes for $\mathbf{X}$, i.e., $\mathbf{x_1}$, $\mathbf{x_2}$, or $\mathbf{x_3}$. The graph $g_1$ generates $\mathbf{x_1}$ and $\mathbf{x_2}$ with probabilities $r$ and $1 - r$ respectively. Likewise, $g_2$ generates $\mathbf{x_2}$ and $\mathbf{x_3}$ with respective probabilities $s$ and $1 - s$. As we can see, $\mathbf{x_1}$ can only be generated by $g_1$ and $\mathbf{x_3}$ can only be the outcome of $g_2$, while $\mathbf{x_2}$ can be generated by both graphs.

We now focus on the scenario with $s = 0$—the general expressions for $I(\mathbf{X};G)$ and the other entropies are obtained in Section II of the Supplementary Information. In this case, only $g_1$ can generate $\mathbf{x_1}$ and $\mathbf{x_2}$, while $g_2$ can only generate $\mathbf{x_3}$. Therefore, we have perfect reconstructability of either graphs, meaning $U(G|\mathbf{X}) = 1$ for any $p$ and $r$, since the outcome of $\mathbf{X}$ tells us immediately which graph generated it. However, $\mathbf{X}$ is imperfectly predictable from $G$ since $U(\mathbf{X}|G) = \frac{\mathcal{H}(p)}{\mathcal{H}(p) + p\mathcal{H}(r)} < 1$ when $0 < p, r < 1$, even though $\mathbf{x_3}$ can be perfectly predicted from $g_2$ as it is its only possible outcome. The remaining entropy, i.e. the second term of the denominator, $p\mathcal{H}(r)$, corresponds to the uncertainty related to whether $g_1$ generates $\mathbf{x_1}$ or $\mathbf{x_2}$.

When $s = 1$, the system is both partially predictable and reconstructible, with $U(\mathbf{X}|G) = 1 - \frac{p\mathcal{H}(r)}{\mathcal{H}(pr)} < 1$ and $U(G|\mathbf{X}) = \frac{\mathcal{H}(pr) - p\mathcal{H}(r)}{\mathcal{H}(pr)} < 1$ for all $0 < p, r < 1$. Both $g_1$ and $g_2$ can generate $\mathbf{x_2}$, but the probability that $g_1$ generates $\mathbf{x_2}$ decreases with $r$. This results in a gradual increase of predictability and reconstructability as $r$ approaches 1, where the systems tend to a one-to-one mapping between the outcomes of $G$ and $\mathbf{X}$.

The intermediate cases when $0 < s < 1$ are also interesting because they give rise to an interval in $r$ where the system becomes less predictable but more reconstructable as $r$ increases, as highlighted by the gray area in Fig. 4c. This happens because, as $r$ increases for a fixed $s$, the growth of entropy of $\mathbf{X}$ dominates $I(\mathbf{X};G)$, resulting in the dual behavior of $U(\mathbf{X}|G)$ and $U(G|\mathbf{X})$.

## $\theta$-duality between predictability and reconstructability

Predictability and reconstructability in dynamics on random graphs offer two perspectives of the same information shared by $G$ and $\mathbf{X}$—two sides of the same coin. However, as we have previously seen with simple examples, predictability, and reconstructability do not necessarily go hand in hand even though they are related: An increasing $U(G|\mathbf{X})$ according to some parameter $\theta$ of the system does not necessarily imply an increase of $U(\mathbf{X}|G)$ and vice versa. Furthermore, a high value of $U(G|\mathbf{X})$ is not tied to a high value of $U(\mathbf{X}|G)$, and conversely, as illustrated in Fig. 1b, c. Indeed, $U(G|\mathbf{X})$ and $U(\mathbf{X}|G)$ can take opposing values, depending on $H(G)$ and $H(\mathbf{X})$, for the same value of $I(\mathbf{X};G)$. This phenomenon can also be observed in the performance of prediction and reconstruction (see Supplementary Note I). In the literature, a hint of the existence of such dual behavior was recently corroborated in ref. [51] for continuous-state deterministic dynamics. The authors showed that high prediction accuracy can be achieved with graphs reconstructed from the very time series they want to predict, even if they are different from the original graph that

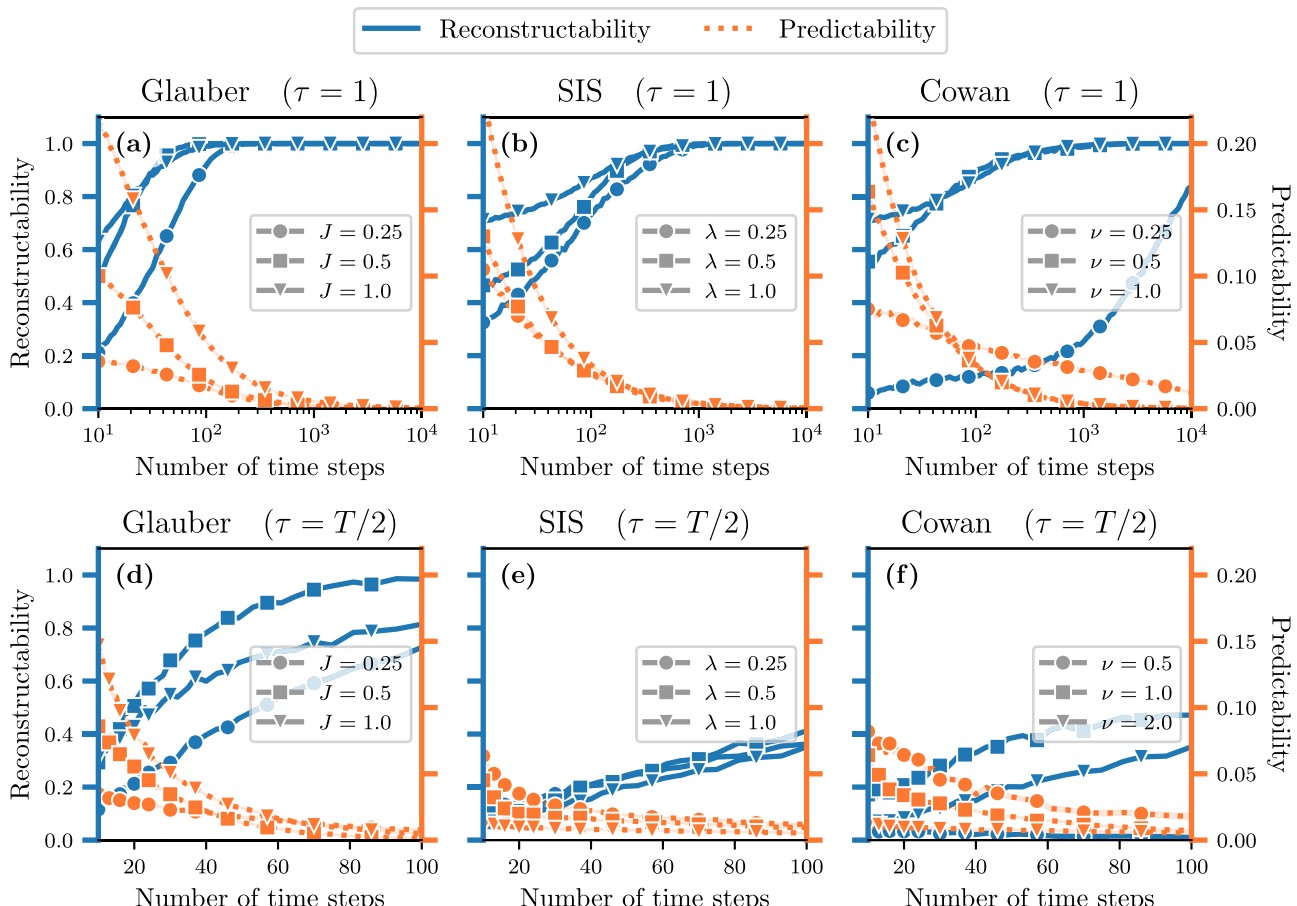

**Fig. 5 | *T*-duality in binary dynamics evolving on small Erdős-Rényi random graphs. a**, **d** Glauber dynamics, **b**, **e** SIS dynamics, and **c**, **f** Cowan dynamics. Each panel shows the reconstructability $U(G\,|\,\mathbf{X}) \in [0,1]$ (blue) and the predictability coefficient $U(\mathbf{X}\,|\,G) \in [0,1]$ (orange) as a function of the number of time steps $T$. We used graphs of $N=5$ vertices and $E=5$ edges, meaning an average degree of $\langle k \rangle = 2$; we fixed $\tau = 1$ in the top row, and $\tau = T/2$ in the bottom row. Each symbol corresponds to the average value measured over 1000 samples. We also show different values of the coupling parameters using different symbols: **a**, **d** $J \in \{\frac{1}{2}, 1, 2\}$ for Glauber, **b**, **e** $\lambda \in \{\frac{1}{2}, 1, 2\}$ for SIS, and **c**, **f** $\nu \in \{\frac{1}{2}, 1, 2\}$ for Cowan.

generated the time series. This phenomenon can be understood through our framework (see Section III of the Supplementary Information) and we now devote the rest of the section to precisely define and characterize the somewhat counterintuitive phenomenon of duality.

We identify a duality when $U(\mathbf{X}\,|\,G)$ and $U(G\,|\,\mathbf{X})$ vary in opposite directions when a parameter, say $\theta$, is changed. More specifically, we say that they are dual with respect to $\theta$, or $\theta$-dual, in an interval $\Theta$ if and only if the signs of their derivative with respect to $\theta$ are different for every $\theta^* \in \Theta$:

$$\left[\frac{\partial U(G\,|\,\mathbf{X})}{\partial \theta}\frac{\partial U(\mathbf{X}\,|\,G)}{\partial \theta}\right]_{\theta=\theta^*} < 0 \,. \tag{5}$$

This criterion formally relies on the existence of regions $\Theta$ where the variations of $U(G\,|\,\mathbf{X})$ and $U(\mathbf{X}\,|\,G)$ with respect to $\theta$ are opposite, regardless of their amplitude (see also "Formal definition of $\theta$-duality" section). We use this criterion to relate the existence of extrema of $U(G\,|\,\mathbf{X})$ and $U(\mathbf{X}\,|\,G)$ with that of regions of $\theta$-duality (see Lemma 1 in "Formal definition of $\theta$-duality" section).

With our intuition being established from simple examples and our precise definition, we are finally ready to state one of the main results of the paper. Recalling that $T$ is the length of process $\mathbf{X}$, we prove that reconstructability and predictability are $T$-dual for a vast class of Markov chains.

**Theorem 1.** Let $\mathbf{X} = (\mathbf{X}_1, \mathbf{X}_2, \cdots, \mathbf{X}_T)$ be a Markov chain of length $T$ whose transition probabilities are conditional to some discrete random variable $G$ that is independent of $T$ and such that $H(\mathbf{X}_{t+1}|\mathbf{X}_t) > 0$ for all $t \in \{1, \ldots, T-1\}$ (i.e., $\mathbf{X}$ is non-deterministic). Moreover, suppose that the state spaces of $\mathbf{X}$ and $G$ are finite, and that $\mathbf{X}$ has a finite nonzero entropy rate and that $G$ has a nonzero entropy. Then there exists a positive constant $\phi$ such that the uncertainty coefficients $U(G|\mathbf{X})$ and $U(\mathbf{X}|G)$ are $T$-dual for all $T \geq \phi$.

The proof of this theorem is in "Proof of the universality of the T-duality" section. It is a consequence of the fact that the mutual information is strictly increasing with $T$—and so is $U(G\,|\,\mathbf{X})$ since $H(G)$ is independent of $T$—whenever the entropy rate of $\mathbf{X}$ is positive. As a result, $U(G\,|\,\mathbf{X})$—and numerator, $I(\mathbf{X}; G)$—stagnates at some point in $T$, while $U(\mathbf{X}\,|\,G)$ keeps decreasing because its denominator increases in an asymptotically linear manner with $T$. We refer to this opposing behavior as a duality between $U(G\,|\,\mathbf{X})$ and $U(\mathbf{X}\,|\,G)$ with respect to $T$, or a $T$-duality for short (not to be confused with target space duality in string theory[85].) When the entropy rate is not well-defined, like for non-stationary processes, the universality of the $T$-duality might not hold, while it remains possible to observe it in localized intervals of $T$.

Figure 5 illustrates the universality of the $T$-duality using the special case of binary Markov chains (i.e., $\Omega = \{0,1\}$, see "Binary Markov chains on graphs" section). These systems are parametrized by their activation $(0 \to 1)$ and deactivation $(1 \to 0)$ probability functions, denoted $\alpha(n_{i,t}, m_{i,t})$ and $\beta(n_{i,t}, m_{i,t})$, respectively. In general, the

**Table 1 | Activation and deactivation probability functions α(n, m) and β(n, m)**

| Dynamics | α (n, m) | β(n, m) | Coupling |
|---|---|---|---|
| Glauber[86] | $\sigma(2J(n-m))$ | $\sigma(2J(m-n))$ | $J$ |
| SIS[5] | $1-\left(1-\frac{\lambda}{\beta}\right)^m$ | $\beta$ | $\lambda$ |
| Cowan[90] | $\sigma(a(\nu m - \mu))$ | $\beta$ | $\nu$ |

For the binary dynamics considered in this study, we show the functional form of α(n, m) and β(n, m), where n corresponds to the number of inactive neighbors whose states are 0, and m corresponds to the number of active neighbors whose states are 1. We define $\sigma(x) = [\exp(-x)+1]^{-1}$ as the logistic function. Some of these parameters are fixed throughout the paper: $\beta = 0.5$ for SIS and Cowan, and $a = 7$ and $\mu = 1$ for Cowan. The coupling parameters ($J$ for Glauber, λ for SIS, and ν for Cowan) are specified in each dynamics. Also, to prevent the SIS dynamics from being completely inactive, we allow the inactive vertices to spontaneously activate with probability $\epsilon = 10^{-3}$ [94].

activation and deactivation functions depends solely on $n_{i,t}$ and $m_{i,t}$, i.e., the number of active and inactive neighbors of vertex $i$ at time $t$. We present multiple examples of binary Markov processes with different origins in Table 1: The Glauber dynamics, the Suspectible-Infectious-Susceptible (SIS) dynamics, and the Cowan dynamics.

The aforementioned Glauber dynamics[86], which have been used to describe the time-reversible evolution of magnetic spins aligning in a crystal, have been tremendously studied because of its critical behavior and its phase transition. Its stationary distribution is given by the Ising model which has found many applications in condensed-matter physics[87] and statistical machine learning[81,88], to name a few. The SIS dynamics is a canonical model in network epidemiology[5] often used for modeling influenza-like disease[89], where periods of immunity after recovery are short. In this model, susceptible (or inactive) vertices get infected by each of their infected (active) first neighbors, with a constant transmission probability, and recover from the disease with a constant recovery probability. The simplicity of the SIS model has allowed for deep mathematical analysis of its absorbing-state phase transition[15,17,20]. Finally, the Cowan dynamics[90] has been proposed to model the neuronal activity in the brain. In this model, quiescent neurons fire if their input current, coming from their firing neighbors, is above a given threshold. Its mean-field approximation[91] reduces to the Wilson-Cowan dynamics[92], one of the most influential models in neuroscience[93]. For each model, we can identify an inactive state—down, susceptible, or quiescent—and an active one—up, infectious, or firing. The corresponding activation and deactivation probabilities are given in Table 1.

Figure 5 numerically supports Theorem 1 and clearly illustrates the $T$-duality for each dynamics, with different values of their parameters and different past lengths τ. We used the Erdős-Rényi model as the random graph on which these dynamics evolve. The support $\mathcal{G}$ is the set of all simple graphs of $N$ vertices with $E$ edges, and

$$P(G) = \left(\binom{N}{2}\atop E\right)^{-1}. \tag{6}$$

Note that, in this example, we consider the well-known Erdős-Rényi model for simplicity (Eq. (6)). Furthermore, we considered very small graphs of size $N = 5$, because the exact evaluation of $I(\mathbf{X}; G)$ is computationally intractable. For larger systems, biased estimators can be designed to bound $I(\mathbf{X}; G)$ as we show in "Estimators of the mutual information" section. We demonstrate the flexibility of our framework with regard to the random graph models by using more sophisticated and data-driven graph models in the following section.

The $T$-duality persists for the past-dependent measures presented in "Past-dependent mutual information" section, as illustrated by the bottom row of Fig. 5, for $\tau = T/2$. However, note that for sufficiently large τ, the duality seems to disappear. We refer to Section VII of

the Supplementary Information for further detail. One can only wonder how many different kinds of parameters can lead to θ-dualities. Maybe some may control the general behavior of the dynamics, and others some aspect of the system structure which, in turn, may also impact the dynamics. In the next section, we investigate those that are related to critical phenomena in complex systems.

**Duality and criticality**

Despite their different nature and range of applications, the three models presented in Table 1 share several properties of interest. For instance, each model has a coupling parameter that controls the influence of the state of the first neighbors on the transition probabilities. They also all feature a phase transition in the infinite size limit whose position is determined by the coupling parameter (see Section IX of the Supplementary Information). We now investigate the influence of criticality over the existence of θ-dualities, where θ is a coupling parameter.

For the Glauber dynamics, this parameter is the coupling constant $J$, which dictates the reduction (increase) in the total energy of a spin configuration when two neighboring spins are parallel (antiparallel). The Glauber dynamics features a continuous phase transition at a critical point $J_c$ between a disordered and an ordered phase, where for $J < J_c$ the spins are disordered resulting in a vanishing magnetization, and for which this magnetization is nonzero when $J > J_c$.

For the SIS dynamics, it is the transmission rate λ that acts as a coupling parameter. Like the Glauber dynamics, the SIS dynamics possesses a continuous phase transition where, when $\lambda < \lambda_c$, the system reaches an absorbing—or inactive—state from which it cannot escape, and an active state, when $\lambda > \lambda_c$, where a nonzero fraction of the vertices remain active over time. It should be emphasized that in our considered version of the SIS dynamics, referring to the system reaching a true absorbing state is not strictly accurate due to the allowance for self-infection $\epsilon$, which enables escape from the completely inactive state. Instead, the system approaches a metastable state with most vertices becoming asymptotically inactive. However, it can be shown that the two-phase transitions are quite similar for small $\epsilon$[94].

The Cowan dynamics can both feature a continuous or a first-order phase transition between an inactive and an active phase depending on the value of slope $a$, for which the coupling parameter is ν, i.e., the potential gain for each firing neighbors. The continuous and first-order phase transitions of the Cowan dynamics are quite different in that the latter is characterized by two thresholds, namely the forward and backward thresholds $\nu_c^b < \nu_c^f$, respectively (see Section IX in the Supplementary Information). Hence, the Cowan dynamics has a first-order phase transition that exhibits a bistable region $\nu \in (\nu_c^b, \nu_c^f)$, where both the inactive and active phases are reachable depending on the initial conditions.

To account for the heterogeneous network structure observed in a wide range of complex systems[1], we simulate the dynamics on the configuration model, a random graph whose—potentially heterogeneous—degree sequence $\boldsymbol{k}$ is fixed and whose support $\mathcal{G}$ corresponds to the set of all loopy multigraphs of degree sequence $\boldsymbol{k}$. The probability of a multigraph $g$ in this ensemble is

$$P(G = g) = \frac{(2E)!!}{(2E)!}\frac{\prod_i k_i!}{\prod_{i<j}M_{ij}!\prod_i M_{ii}!!}, \tag{7}$$

where $M_{ij}$ counts the number of edges connecting vertices $i$ and $j$ in the multigraph $g$ and $2E = \sum_i k_i$ is the number of half-edges in $g$. Like the Erdős-Rényi model, the configuration model fixes the number of edges, but also fixes the degree sequence $\boldsymbol{k} = (k_1, \cdots, k_N)$.

Figure 6 shows the predictability and reconstructability, as estimated by the MF estimator, of the three dynamics evolving on instances drawn from the configuration model. The top row shows the

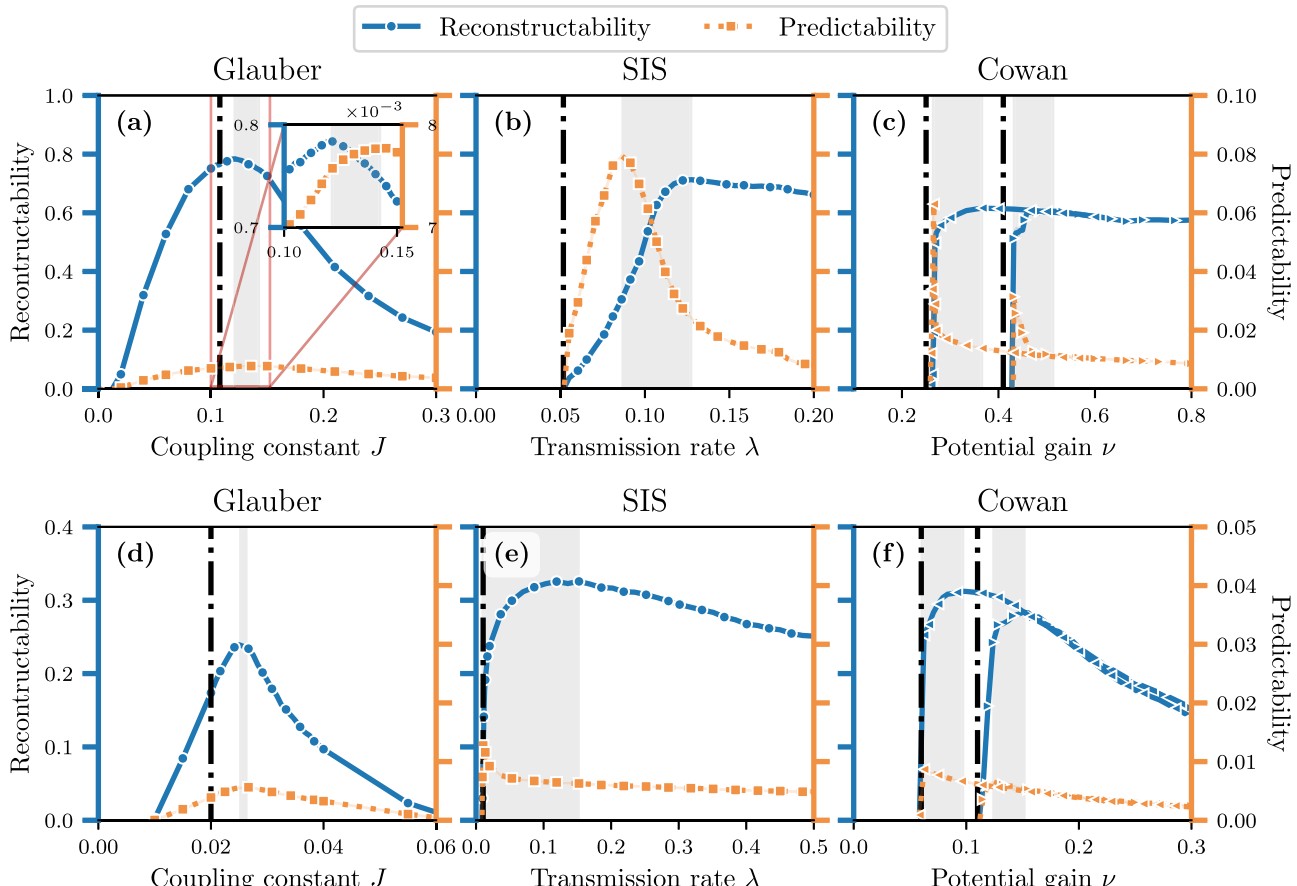

**Fig. 6 | Dynamics evolving on configuration model graphs. a, d** Glauber dynamics, **b, e** SIS dynamics, and **c, f** Cowan dynamics. We used the configuration model (see Eq. (7)) to generate multigraphs of varying sizes and degree distributions. In the top row, we generated multigraphs with geometric degree distribution of size $N = 1000$ and with $M = 2500$ edges (see Fig. 7a). In the bottom row, we used the degree distribution of real networks: **d** Little Rock Lake food web[95], **e** European airline route network[96], **f** C. Elegans neural network[97]. The parameters used to generate the time series are the same in the top and bottom panels (see Table 1), except in **f** the time series length is $T = 5000$ while in the others $T = 2000$. Similar to

Fig. 5, $U(G \mid \mathbf{X})$ is shown in blue (left axis) and $U(\mathbf{X} \mid G)$ is shown in orange (right axis). We show, for each dynamics, the uncertainty coefficients as a function of the coupling parameter: $J$ for Glauber, $\lambda$ for SIS, and $\nu$ for Cowan. Each shaded area indicates a range of couplings over which duality was observed. The vertical dotted-dashed lines correspond to the phase transition thresholds of each dynamics, which are estimated from Monte Carlo simulations (see Section IX of the Supplementary Information). For the Cowan dynamics, the forward and backward branches are shown with their corresponding thresholds and dual regions (see main text).

results when using a synthetic degree sequence obtained from a geometric degree distribution, while for the bottom row, degree sequences from different real networks are used for each dynamics. These distributions are shown in Fig. 7. We used the Little Rock Lake food web[95] (as in ref. 40) jointly with the Glauber dynamics to simulate a simplification of the interaction between species. In the case of the SIS dynamics, we considered the European airline network[96] to mimic the spread of an epidemic. Finally, to simulate the neural activity of the Cowan dynamics, we used the C. Elegans neural network[97].

First, the results of Fig. 6 show a meaningful comparison between the dynamics for different types of structures. For example, on the one hand, the Glauber dynamics is globally less predictable than the other two, since its predictability coefficient is overall smaller. In other words, the knowledge of a graph $g$ provides less information about $\mathbf{X}$ in the Glauber dynamics in comparison with the others, relative to the total amount of information needed to predict $\mathbf{X}$. This is related to the time reversibility of the Glauber dynamics, which allows any vertex to transition from the inactive to the active state (and vice versa) with nonzero probability, at any time, effectively making the Glauber dynamics more random than the others—i.e. $H(\mathbf{X})$ is greater for Glauber than the other processes. On the other hand, the SIS and Cowan dynamics are shown as practically unpredictable and unreconstructable when their coupling parameter is

below their respective critical point. This precisely occurs in the inactive phase, where the system rapidly reaches the inactive state and no mutual information can be generated. By contrast, the Glauber dynamics does not reach an inactive state below its critical point, which explains the gradual increase in predictability and reconstructability in that region.

Several additional observations are worth making. All dynamics exhibit maxima for $U(\mathbf{X} \mid G)$ and $U(G \mid \mathbf{X})$, which delineate a region of duality illustrated by the shaded areas (two for Cowan, that is one for each branch). These regions are close to, but systematically above, their respective phase transition thresholds, regardless of type of degree sequence. A similar phenomenon in spin dynamics on non-random lattices has been reported by previous works[79,80], in which the information transmission rate between spins—a measure akin to $I(\mathbf{X}; G)$ —is maximized above the critical point. Our numerical results are consistent with theirs, and suggest that their findings regarding near-critical systems even apply beyond spin dynamics on fixed lattices, to other types of processes on more heterogeneous and random structures.

## Discussion
In this work, we used information theory to characterize the structure-function relationship with mutual information. We showed how

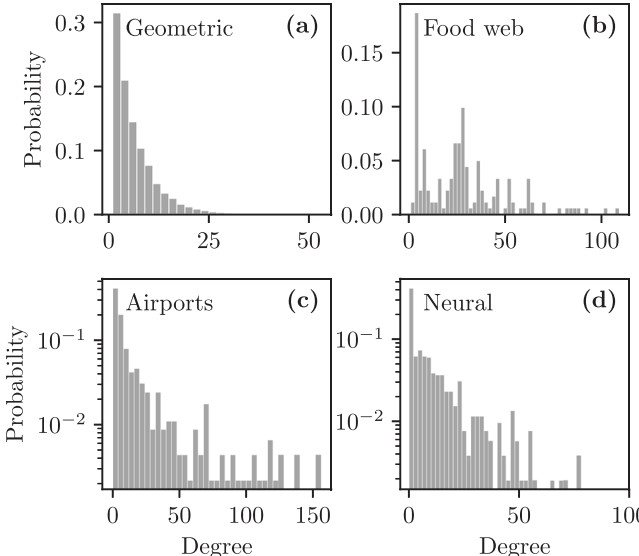

**Fig. 7 | Degree distributions of the graphs used in Fig. 6. a** multigraphs with geometric degree distribution $\rho(k) = (1-p)p^k$ where $p = 5/6$, with $N = 1000$ nodes and $M = 2500$ edges, **b** Little Rock Lake food web[95], **c** European airline route network[96], **d** C. Elegans neural network[97]. See "Real networks" section for further details about the graphs.

mutual information is a natural starting point to define both predictability and reconstructability in dynamics on networks, and even how it explains the performance accuracy of prediction and reconstruction algorithms. In turn, we demonstrated how prediction and reconstruction in complex systems are intrinsically related. Our approach is quite general, allowing the exploration of different configurations of dynamics on networks of the form $G \to X$, thus varying the nature of the process itself as well as the random graph on which it evolves. Our framework could be extended to adaptive systems[98–101] where both $X$ and $G$ influence each other (i.e., $X \leftrightarrow G$). The relationship between $X$ and $G$ could also go the other way around: A system in which $X$ generates a graph $G$ (i.e., $X \to G$). Hyperbolic graphs[102,103] fall into this category, where $X$ represents a set of coordinates, and our framework could be extended to quantifying the feasibility of network geometry inference[104–106].

We exposed various examples where our measures can be computed analytically and found efficient ways to estimate them numerically when needed, thus allowing thorough investigation of large systems. More work on this front is required, however, since the evaluation of these estimators remains quite computationally costly. It would be worth investigating dimension-reduction methods[11,13,14] and approximate master equations[18,107], among others, for obtaining more efficient and reliable approximations of $I(\mathbf{X}; G)$, $U(\mathbf{X} \mid G)$ and $U(G \mid \mathbf{X})$.

Central to our findings is the peculiar discovery that predictability and reconstructability are not only related, but sometimes dual to one another. We found many examples of this duality in systems of increasing complexity, while we also emphasized that its universality is limited to certain circumstances. One of those circumstances occurs when we change the length of the processes, for which we mathematically proved the existence of duality. We also presented numerical evidence of duality near-critical points in three different dynamics on real networks. These findings generalize and formalize—while being consistent with—previous works[79,80] and suggest that the reconstructability-predictability duality with respect to order parameters is closely linked to the criticality in these systems.

From a practical perspective, the existence of such a $\theta$-duality can be critical to network modeling applications, since it also suggests a predictability-reconstructability trade-off. On one hand, by choosing the parameter $\theta$, we can minimize the uncertainty of the reconstructed structure, but this may result in a structure that is less informative regarding the dynamics. On the other hand, we can consider the reverse case, where the process is maximally influenced by the inferred structure, whose uncertainty is nevertheless not minimized. Analogous to the position-momentum duality in the Heisenberg uncertainty principle of quantum mechanics, the predictability-reconstructability duality must be accounted for in our network models if we are to disentangle complex systems.

## Methods

### Binary Markov chains on graphs

The models used throughout the paper are for the most part Markov chains $\boldsymbol{X} = (\boldsymbol{X}_1, \boldsymbol{X}_2, ..., \boldsymbol{X}_T)$, that are governed by a conditional probability $P(\boldsymbol{X}|G)$ that can be factored as follows:

$$P(\mathbf{X} \mid G) = P(\mathbf{X}_1) \prod_{t=1}^{T-1} P(\mathbf{X}_{t+1} \mid \mathbf{X}_t, G). \tag{8}$$

The probability $P(\boldsymbol{X}_{t+1}|\boldsymbol{X}_t, G)$ is the global transition probability from state $\boldsymbol{X}_t$ to state $\boldsymbol{X}_{t+1}$, and $P(\boldsymbol{X}_1)$ represents the probability distribution of the initial conditions, which is independent of $G$ in our case. More specifically, we assume that $\boldsymbol{X}_i$ is a random binary vector of size $N$, and that the global transition probability can be factored in terms of local transition probabilities as follows:

$$P(\mathbf{X}_{t+1} \mid \mathbf{X}_t, G) = \prod_{i=1}^{N} \left\{ \left[\alpha(n_{i,t}, m_{i,t})\right]^{(1-X_{i,t})X_{i,t+1}} \left[1 - \alpha(n_{i,t}, m_{i,t})\right]^{(1-X_{i,t})(1-X_{i,t+1})} \right.$$
$$\left. \left[\beta(n_{i,t}, m_{i,t})\right]^{X_{i,t}(1-X_{i,t+1})} \left[1 - \beta(n_{i,t}, m_{i,t})\right]^{X_{i,t}X_{i,t+1}} \right\}. \tag{9}$$

As mentioned in "$\theta$-duality between predictability and reconstructability" section, the functions $\alpha$ and $\beta$ correspond to the activation and deactivation probabilities. In the general case, they are dependent on the number of active neighbors $m_i$, and inactive neighbors $n_i$ of a node $i$ such that $m_i + n_i = k_i$ where $k_i$ is the degree of this node.

### Performance of prediction and reconstruction algorithms

To substantiate our claim about the interpretation of $I(\mathbf{X}; G)$, we used different prediction and reconstruction algorithms and compared in Fig. 2 their performance with $I(\mathbf{X}; G)$. In this section, we elaborate on this analysis.

**Prediction algorithms.** The prediction algorithms used in Fig. 2 correspond to Markov models that predicts a transition—activation and deactivation—probability matrix $P$, where $P_{i,t}$ corresponds to the probability that node $i$ at time $t$ transition to the active state in the next time step. To make the comparison with $I(\mathbf{X}; G)$, we compare the transition probability matrix $P^*$ of the true model—in the case of Fig. 2, the Glauber dynamics where the entries of $P^*$ are given by the activation $\alpha$ and deactivation $\beta$ probabilities (see Table 1)—with those predicted by models learned from time series generated by the Glauber dynamics. These models are trained with 100 concatenated time series, each generated using a different graph sampled from the Erdős-Rényi model. The models are then trained to predict the time series without the knowledge of the structure. The input of these models is the complete state of the system at time $t$, i.e., $\boldsymbol{X}_t$, and the output is a vector $\hat{P}_t = (\hat{P}_{1,t}, \cdots, \hat{P}_{N,t})$, where $\hat{P}_{i,t}$ is the predicted probability that node $i$ transition to the active state at time $t$. We use the mean absolute

error (MAE) between $P^*$ and $\hat{P}$ to compare them, i.e.,

$$\text{MAE}\left(P^*, \hat{P}\right) = \frac{1}{NT} \sum_{i=1}^{N} \sum_{t=1}^{T} \left| P^*_{i,t} - \hat{P}_{i,t} \right|. \tag{10}$$

In doing so, the MAE quantifies the difference between a graph-dependent model and a graph-independent one, which highlights the importance of $G$ over the prediction of $\mathbf{X}$, which is a proxy of $I(\mathbf{X}; G)$.

We consider two graph-independent prediction models: a logistic regression model and a multilayer perceptron (MLP). In both models, the predicted transition probabilities at time $t$ are given by

$$\hat{P}_t = \frac{1}{e^{-f(\mathbf{X}_t)} + 1}, \tag{11}$$

where $f(\mathbf{X}_t)$ is a learnable function, that is linear for the logistic regression model, i.e.

$$f_{\text{logistic}}(\mathbf{X}_t) = \mathbf{A}\mathbf{X}_t + \mathbf{b}, \tag{12}$$

and non-linear for the MLP:

$$f_{\text{MLP}}(\mathbf{X}_t) = \text{ReLU}\left[ \mathbf{W}_2 \text{ReLU}(\mathbf{W}_1 \mathbf{X}_t + \mathbf{b}_1) + \mathbf{b}_2 \right], \tag{13}$$

such that

$$\text{ReLU}(x) = \begin{cases} x & \text{if } x > 0, \\ 0 & \text{otherwise} \end{cases}. \tag{14}$$

The weight matrices $\mathbf{A}$, $\mathbf{W}_1$, and $\mathbf{W}_2$, and bias vectors $\mathbf{b}$, $\mathbf{b}_1$, and $\mathbf{b}_2$, are learned via stochastic gradient descent using a cross-entropy loss.

**Reconstruction algorithms.** In Fig. 2, we also illustrated the relationship between the performance of reconstruction algorithms and $I(\mathbf{X}; G)$. These algorithms are given the time series and they compute a score matrix $S$, such that $S_{ij}$ for each pair of nodes $(i, j)$ correlates with a probability that an edge exists between them. For the correlation matrix method[31], this score is simply the correlation coefficient:

$$S_{ij} = \frac{C_{ij}}{\sigma_i \sigma_j}, \quad C_{ij} = \frac{1}{T} \sum_{t=1}^{T} (X_{i,t} - \bar{X}_i)(X_{j,t} - \bar{X}_j) \tag{15}$$

where $\bar{X}_i = \frac{1}{T} \sum_{t=1}^{T} X_{i,t}$ and $\sigma_i = \frac{1}{T} \sum_{t=1}^{T} (X_{i,t} - \bar{X}_i)^2$. In the Granger causality method[32], we compare via a F-test the prediction of the time series of a single node $i$ using a linear auto-regressive model, with another auto-regressive model that includes the time series of node $j$. Then, the test determines if the models error are similar or different by computing the following F-statistic:

$$S_{ij} = \frac{\Sigma_{ij}}{\Sigma_i}, \tag{16}$$

where $\Sigma_i$ is the error variance of the auto-regressive model of $i$, and $\Sigma_{ij}$ is the error variance of the other model that also includes $j$. Finally, in the transfer entropy method[33], the score is given by the transfer entropy from the time series of $j$ to the time series of $i$:

$$S_{ij} = T_{X_j \to X_i} \tag{17}$$

where

$$T_{X_j \to X_i} = H(X_{i,t} | X_{i,t-1}) - H(X_{i,t} | X_{i,t-1}, X_{j,t-1}) \tag{18}$$

The entropies involved in the computation of $T_{X_j \to X_i}$ are evaluated using the maximum likelihood estimators of the probabilities

$P(X_{i,t} | X_{i,t-1})$ and $P(X_{i,t} | X_{i,t-1}, X_{j,t-1})$, estimated from the time series itself.

We quantify the accuracy of the reconstruction using the area under the curve (AUC) of the receiver operating characteristic (ROC) curve. This curve is obtained by comparing the true positive rate with the false positive rate, for different thresholds $\phi \in [\min\{S\}, \max\{S\}]$. The AUC, being the integral of that curve, therefore represents the probability that the score matrix $S$ classifies correctly a node pair connected by an edge.

## Formal definition of $\theta$-duality

In what follows, we define the duality between predictability and reconstructability by taking a more general stance: Instead of considering a stochastic process $\mathbf{X}$ evolving on a random graph $G$, we let $G$ be any discrete random variable conditioning the probability of $\mathbf{X}$. First, we define the local duality of the uncertainty coefficients. The latter are considered as continuously differentiable functions with respect to a parameter $\theta$ whose domain is some non-empty interval of the real line.

**Definition 1.** (Local duality). The uncertainty coefficients $U(\mathbf{X} | G)$ and $U(G | \mathbf{X})$ are locally dual with respect to $\theta$ at $\theta = \theta^*$ if and only if

$$\left[ \frac{\partial U(\mathbf{X} | G)}{\partial \theta} \frac{\partial U(G | \mathbf{X})}{\partial \theta} \right]_{\theta = \theta^*} < 0. \tag{19}$$

The definition of the $\theta$-duality, a global property, follows that of the local duality.

**Definition 2.** ($\theta$-Duality). The uncertainty coefficients $U(\mathbf{X} | G)$ and $U(G | \mathbf{X})$ are dual with respect to $\theta$, or $\theta$-dual, in the interval $\Theta$ if and only if they are locally dual for all values of $\theta^*$ in $\Theta$.

From these definitions, we relate the presence of extrema of $U(\mathbf{X} | G)$ and $U(G | \mathbf{X})$ with the existence of a $\theta$-duality.

**Lemma 1.** ($\theta$-duality between extrema). Let $\Theta$ be a non-empty sub-interval of the variable $\theta$ whose one endpoint is a local extremum of $U(\mathbf{X} | G)$ and the other, a local extremum of $U(G | \mathbf{X})$. Moreover, suppose that $U(\mathbf{X} | G)$ and $U(G | \mathbf{X})$ do not have critical points in $\Theta$. Then the extrema points delineate a region of $\theta$-duality if and only if they are both maxima (or both minima).

The proof of this lemma is available in Supplementary Note IV.

## Proof of the universality of the $T$-duality

In what follows, we prove Theorem 1, that shows the universality of the $T$-duality, where $T$ is the number of steps in the process $\mathbf{X}$. We make use of the two following lemmas, that are proved in Supplementary Information (Notes V and VI), regarding the monotonicity of $I(\mathbf{X}; G)$ with respect to $T$ and the existence of continuous extensions of $U(\mathbf{X} | G)$ and $U(G | \mathbf{X})$, that will allow us to apply the Definition 1 involving derivatives.

**Lemma 2.** (Monotonicity of mutual information information with $T$). Let $\mathbf{X} = (\mathbf{X}_1, \mathbf{X}_2, \cdots, \mathbf{X}_T)$ be a Markov chain of length $T$ whose transition probabilities are conditional to some discrete random variable $G$ that is independent of $T$ and such that $H(\mathbf{X}_{t+1} | \mathbf{X}_t) > 0$ for all $t \in \{1, \ldots, T-1\}$. Suppose moreover that the state spaces of $\mathbf{X}$ and $G$ are finite. Then the mutual information $I(\mathbf{X}; G)$ is nonzero and monotonically increasing with $T \in \mathbb{Z}_+$.

**Lemma 3.** (Continuous extension of uncertainty coefficients with $T$). Let $\mathbf{X} = (\mathbf{X}_1, \mathbf{X}_2, \cdots, \mathbf{X}_T)$ and $G$ respectively be a Markov chain and a discrete random variable as in Lemma 2. Then the uncertainty coefficients $U(G | \mathbf{X})$ and $U(\mathbf{X} | G)$, interpreted as functions of $T \in \mathbb{Z}_+$, can be uniquely generalized to functions, respectively $f(T)$ and $g(T)$, that are

holomorphic for all $T \in \mathbb{C}$, and thus real analytic for all $T \in \mathbb{R}_+$. Moreover, $H(\mathbf{X})$ can be extended to a function $h(T)$ that is analytic for all $T \in \mathbb{R}_+$ except where $f(T) = 0$.

Next, we prove Theorem 1.

**Proof.** According to Lemma 3, the quantities $U(\mathbf{X}|G)$, $U(G|\mathbf{X})$, and $H(\mathbf{X})$, which were originally defined as real functions of $T \in \mathbb{Z}_+$, have unique analytic extensions on the positive real axis, i.e., $T \in \mathbb{R}_+$. This allows us to treat $U(\mathbf{X}|G)$, $U(G|\mathbf{X})$, and $H(\mathbf{X})$ as continuously differentiable functions with respect to $T$, where $U(G|\mathbf{X}) = \frac{I(\mathbf{X};G)}{H(G)}$ and $H(\mathbf{X})$ are also monotone.

Now, by hypothesis, the entropy rate of the Markov chain $\mathbf{X}$, $R := \lim_{T \to \infty} \frac{H(\mathbf{X})}{T}$, is well defined and nonzero. Hence, $H(\mathbf{X}) \sim RT$, i.e., $H(\mathbf{X})$ is positive and asymptotically linearly increasing with $T$. Moreover, since $G$ is independent of $T$ and $I(\mathbf{X}; G) > 0$, it follows that $I(\mathbf{X}; G)$ is monotonically increasing with respect to $T$ by Lemma 2. As a result, $U(G|\mathbf{X}) = \frac{I(\mathbf{X};G)}{H(G)}$ is also monotonically increasing, since its denominator is independent of $T$, by assumption. This translates to the strict inequality $\frac{\partial U(G|\mathbf{X})}{\partial T} > 0$. If there exists a $T$-duality, i.e., there is a domain of $T$ where Eq. (19) is true, then $U(\mathbf{X}|G)$ must be monotonically decreasing with $T$—or $\frac{\partial U(\mathbf{X}|G)}{\partial T} < 0$—in that domain. To prove this, note that we can relate the two uncertainty coefficients using

$$H(\mathbf{X}) = \frac{H(G) U(G|\mathbf{X})}{U(\mathbf{X}|G)} . \tag{20}$$

This leads to the following differential equation

$$\frac{\partial}{\partial T}\left[\log U(\mathbf{X}|G)\right] = \frac{\partial}{\partial T}\left[\log U(G|\mathbf{X})\right] - \frac{\partial}{\partial T}\left[\log H(\mathbf{X})\right], \tag{21}$$

where we used the fact that $\frac{\partial H(G)}{\partial T} = 0$. Hence, to show that $U(\mathbf{X}|G)$ is monotonically decreasing with $T$, the following inequality must hold

$$\frac{\partial}{\partial T}\left[\log U(\mathbf{X}|G)\right] < \frac{\partial}{\partial T}\left[\log H(\mathbf{X})\right] . \tag{22}$$

Suppose for a moment that $U(\mathbf{X}|G)$ is in fact increasing, such that Eq. (22) is false. This will eventually give rise to a contradiction. Let $g(T) := U(G|\mathbf{X})$ and $h(T) := H(\mathbf{X})$ be continuous functions of $T$ such that their derivative with respect to $T$ are respectively given by $g'(\tau) := \frac{\partial g(T)}{\partial T}\big|_{T=\tau}$ and $h'(\tau) := \frac{\partial h(T)}{\partial T}\big|_{T=\tau}$. Note that $0 < f(\tau) \le 1$ and $h(\tau) > 0$ for all $\tau \in \mathbb{R}_+$. If Eq. (22) is false, then

$$(\log g(T))' \ge (\log h(T))' . \tag{23}$$

Using Grönwall's inequality[108], Theorem 1.2.1, we get

$$\frac{g(T)}{g(a)} \ge \frac{h(T)}{h(a)} , \quad 0 < a < T . \tag{24}$$

So far, we have established that $h(T) = H(\mathbf{X}) \sim RT$ and that $U(G|\mathbf{X})$ is monotonically increasing. We have also proved that if $U(\mathbf{X}|G)$ is not monotonically decreasing with $T$, then inequality (24) is satisfied. However, the latter inequality and $h(T) \sim RT$ readily imply that $g(T)$ belongs to the class $\Omega(T)$, which is the set of all $\tilde{g}(T)$ such that there exist positive constants, $S$ and $T^*$, for which $\tilde{g}(T) \ge ST$ for all $T \ge T^*$ (i.e., Knuth's Big Omega[109]).

Two cases must be considered. First, if $ST^* > 1$, then $\tilde{g}(T) \ge ST^* > 1$, which is in direct contradiction with $g(T) \le 1$ whenever $T \ge T^*$. Second, if $ST^* \le 1$, then choose $T^{**} > S^{-1} \ge T^*$, so that $\tilde{g}(T) \ge ST^{**} > 1$ for all $T \ge T^{**}$. This again contradicts the inequality $g(T) \le 1$ whenever $T \ge T^{**}$. As a result, inequality (24) cannot be satisfied when $T \ge \phi$, with $\phi = \max\{T^*, T^{**}\}$.

We thus conclude that $U(\mathbf{X}|G)$ is monotonically decreasing for all $T \ge \phi$. Therefore, $U(G|\mathbf{X})$ and $U(\mathbf{X}|G)$ are $T$-dual in the interval $[\phi, \infty)$. □

## Estimators of the mutual information

The mutual information $I(\mathbf{X}; G)$ is generally intractable. Its intractability stems from the evaluation of the evidence probability, which is defined by the following equation:

$$P(\mathbf{X} = \mathbf{x}) = \sum_{g \in \mathcal{G}} P(G = g) P(\mathbf{X} = \mathbf{x}|G = g) . \tag{25}$$

Indeed, this sum potentially counts a number of terms which grows exponentially with the number of vertices $N$ in the random graph. More specifically, the evidence probability appears in two entropy terms needed to compute the mutual information, namely the marginal entropy $H(\mathbf{X}) = -\langle \log P(\mathbf{X}) \rangle$ and the reconstruction entropy $H(G|\mathbf{X}) = -\langle \log \frac{P(G)P(\mathbf{X}|G)}{P(\mathbf{X})} \rangle$, where $\langle f(Y) \rangle$ denotes the expectation of $f(Y)$. Fortunately, the evidence probability, and in turn the mutual information, can be estimated efficiently using Monte Carlo techniques, which we present in this section.

**Graph enumeration approach.** For sufficiently small random graphs ($N \le 5$), the evidence probability can be efficiently computed by enumerating all graphs of $\mathcal{G}$ and by adding explicitly each term of Eq. (25). Then, we can estimate the mutual information by sampling $M$ graphs $\{g^{(m)}\}_{m=1..M}$, followed by $M$ time series $\{\mathbf{x}^{(m)}\}_{m=1..M}$—such that $\mathbf{x}^{(m)}$ is generated with $g^{(m)}$—, and by computing the following arithmetic average:

$$I(\mathbf{X}; G) \simeq \frac{1}{M} \sum_{m=1}^{M} \log P(\mathbf{X} = \mathbf{x}^{(m)}|G = g^{(m)}) - \log P(\mathbf{X} = \mathbf{x}^{(m)}) . \tag{26}$$

The variance of this estimator scales with the inverse of $\sqrt{M}$. In Fig. 5, we used this estimator to compute the mutual information, where $M = 1000$.

**Variational mean-field approximation.** In this approach, we estimate the posterior probability instead of the evidence probability. According to Bayes' theorem, the posterior probability is

$$P(G|\mathbf{X}) = \frac{P(G)P(\mathbf{X}|G)}{P(\mathbf{X})} . \tag{27}$$

Behind this estimator is a variational mean-field (MF) approximation that assumes the conditional independence of the edges. For simple graphs, the MF posterior is

$$P_{\mathrm{MF}}(G|\mathbf{X}) = \prod_{i \le j} \left[\pi_{ij}(\mathbf{X})\right]^{A_{ij}} \left[1 - \pi_{ij}(\mathbf{X})\right]^{1 - A_{ij}}, \tag{28}$$

where $\pi_{ij}(\mathbf{X}) := P(A_{ij} = 1|\mathbf{X})$ is the marginal conditional probability of the existence of the edge $(i, j)$ given $\mathbf{X}$. For multigraphs, a similar expression can be obtained, but instead involves a probability $\pi_{ij}(m|\mathbf{X}) := P(M_{ij} = m|\mathbf{X})$ that there are $m$ multiedges between $i$ and $j$. In this case, the MF posterior becomes

$$P_{\mathrm{MF}}(G|\mathbf{X}) = \prod_{i \le j} \prod_{m=0}^{\infty} \left[\pi_{ij}(m|\mathbf{X})\right]^{\delta_{m,M_{ij}}}, \tag{29}$$

where $\delta_{x,y}$ is the Kronecker delta. The MF approximation allows to compute a lower bound of the true posterior entropy, such that

$$H(G|\mathbf{X}) \ge -\langle \log P_{\mathrm{MF}}(G|\mathbf{X}) \rangle, \tag{30}$$

as a consequence of the conditional independence between the edges[82], Theorem 2.6.5. Using the MF approximation and a strategy similar to the exact estimator, we compute the MF estimator of the mutual information as follows:

$$I(G \mid \mathbf{X}) \geq \frac{1}{M} \sum_{m=1}^{M} \left[ \log P_{\mathrm{MF}}\left(G = g^{(m)} \mid \mathbf{X} = \mathbf{x}^{(m)}\right) - \log P\left(G = g^{(m)}\right) \right]. \quad (31)$$

To compute $P_{\mathrm{MF}}\left(G = g^{(m)} \mid \mathbf{X} = \mathbf{x}^{(m)}\right)$, we sample a set $\mathcal{Q}^{(m)} := \left\{ g_1^{(m)}, \cdots, g_Q^{(m)} \right\}$ of $Q$ graphs from the posterior distribution $P(G \mid \mathbf{X} = \mathbf{x}^{(m)})$. Then, we estimate the probabilities $\pi_{ij}(\mathbf{X}) \simeq \frac{n_{ij}^{(m)}}{Q}$ using their corresponding maximum likelihood estimate, where $n_{ij}^{(m)}$ is the number of times the edge $(i, j)$ is seen in $\mathcal{Q}^{(m)}$. An analogous maximum likelihood estimate is made in the multigraph case, where $\pi_{ij}(\omega \mid \mathbf{X}) \simeq \frac{n_{ij;\omega}^{(m)}}{K}$ and $n_{ij;\omega}^{(m)}$ counts the number of times there were $\omega$ multiedges between $i$ and $j$ in $\mathcal{Q}^{(m)}$. This estimator is a lower bound of the mutual information—a consequence of Eq. (30). Hence, it is biased, and the extent of this bias is dependent on the quality of the conditional independence assumption with respect to the true random graph. Note that the MF estimator can yield negative estimates of the mutual information (see Section VIII of the Supplementary Information).

In Fig. 6, we fix the number of graphs sampled from the posterior distribution to $Q = 1000$, and propose $5N$ moves between each sample (see also "Markov chain Monte Carlo algorithm" section for more detail).

## Markov chain Monte Carlo algorithm

To sample from the posterior distribution, we use a Markov chain Monte Carlo (MCMC) algorithm where, starting from a graph $g$, we propose a move to graph $g'$, according to a proposition probability $P(G' = g' \mid G = g)$, and accept it with the Metropolis-Hastings probability:

$$\min \left( 1, e^{-\log \Delta} \frac{P(G' = g \mid G = g')}{P(G' = g' \mid G = g)} \right), \quad (32)$$

where $\Delta = \frac{P(G = g') P(\mathbf{X} = \mathbf{x} \mid G = g')}{P(G = g) P(\mathbf{X} = \mathbf{x} \mid G = g)}$ is the ratio between the joint probability of the two graphs with the time series $\mathbf{X}$. This ratio can be computed efficiently in $\mathcal{O}(T)$, by keeping in memory $n_{i,t}$, the number of inactive neighbors, and $m_{i,t}$, a number of active neighbors, for each vertex $i$ at each time $t$ (see ref. 40). Equation (32) allows to sample from the posterior distribution $P(G \mid \mathbf{X})$ without the requirement to compute the intractable normalization constant $P(\mathbf{X})$. We collect graph samples at every $N\delta$ move, where we fix $\delta = 5$ in all experiments.

We consider two types of random graphs with different constraints: The Erdős-Rényi model and the configuration model. Hence, we need two different sampling propositions to apply our MCMC algorithm, that is one for each model. We assume that the support of the Erdős-Rényi model is the set of all simple graphs of $N$ vertices with $E$ edges. In this case, we consider a hinge flip move, where an edge $(i, j)$ is sampled uniformly from the edge set of the graph $G$ and a vertex $k$ is sampled uniformly from its vertex set. Then, with probability $\frac{1}{2}$, we rewire edge $(i, j)$ by either selecting $i$ or $j$ to connect with $k$. Note that, because we consider the support $\mathcal{G}$ of $G$ to be a space of simple graphs, all moves resulting in the addition of a self-loop or a multiedges are rejected with probability 1. As a result, the proposition probability is the same for any move from $g$ to $g'$:

$$P(G' = g' \mid G = g) = \frac{1}{EN} \Rightarrow \frac{P(G' = g \mid G = g')}{P(G' = g' \mid G = g)} = 1. \quad (33)$$

For the configuration model, we assume that the support is the set of all loopy multigraphs of $N$ vertices whose degree sequence is $\mathbf{k}$. In this case, we propose double-edge swap moves according to the prescription of ref. 110. We refer to it for further details.

## Real networks

In this section, we present the real networks used in the bottom panels of Fig. 6. The networks have been downloaded from the Netzschleuder network catalog[111].

**Little Rock Lake food web.** The Little Rock Lake food web[95] is composed of $N = 183$ nodes and $M = 2\,494$ edges, where nodes represent taxa (like species) found in Little Rock Lake in Wisconsin, and edges represent feeding patterns between two taxa. As presented in ref. 95, this network is directed, but for the purpose of our paper we reciprocated all edges. Also, note that the Glauber dynamics, which we used jointly with the Little Rock Lake food web in Fig. 6, was also used in ref. 40 to simulate a simplified interaction between the taxa.

**European airline route network.** The European airline route network[96] is a multiplex network composed of $N = 450$ and $M = 3\,588$ edges, where nodes represent airports and edges are routes between them. These edges have different types, encoding the different airlines. In our paper, we do not make any distinction between the edge types for simplicity.

**C. Elegans neural network.** The C. Elegans neural network[97] used in Fig. 6 is an undirected network of $N = 514$ and $M = 2363$ edges representing the neural network of male C. Elegans worms. The nodes are neurons and edges represent when there are gap junctions between neurons.

## Reporting summary

Further information on research design is available in the Nature Portfolio Reporting Summary linked to this article.

## Data availability

The real network data used in the paper were downloaded from the network repository Netzschleuder[111]. The details are given in "Real networks" section.

## Code availability

The *Python* code used to generate the results of the paper is available on GitHub[112].

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

## Acknowledgements

We are grateful to Guillaume St-Onge and Vincent Painchaud for useful comments, and to Simon Lizotte and François Thibault for their help in designing the software. This work was supported by the Fonds de recherche du Québec – Nature et technologies (V.T.), the Conseil de recherches en sciences naturelles et en génie du Canada (C.M., A.A., P.D.), the Sentinelle Nord program of Université Laval, funded by the Fonds d'excellence en recherche Apogée Canada (C.M., A.A., P.D.), and the Fonds d'accélération des collaboration en santé du Québec – Alliance Neuro-CERVO (A.A., P.D.). We acknowledge Calcul Québec and Digital Research Alliance of Canada for their technical support and computing infrastructures.

## Author contributions

C.M., V.T., A.A. and P.D. developed the framework and wrote the paper. C.M., V.T. and P.D. wrote the mathematical proofs. C.M. performed the numerical analysis.

## Competing interests

The authors declare no competing interests.
