## [Peer Review File · Nature Communications]

REVIEWER COMMENTS

Reviewer #1 (Remarks to the Author):

The authors have proposed an information theoretic approach to the dependence between the process X on a graph and the structure of that graph G . The manuscript is well written and the story seems compelling at first glance!

Nevertheless, there are serious concerns. The authors have interpreted the mutual information $I(X,G)$ in eq. 1 (1a and 1b) in terms of the “predictability” and “reconstructability”. I think that the authors overaccentuate the meaning of “predictability” and “reconstructability”: how does mutual entropy of X and G translates to “prediction” and “reconstructability”; mathematically, there is only a probabilistic conditional dependence between G and X and vice versa. Instead of predictability, we may talk e. g. about “sensitivity” of X upon G (and vice versa), which is more correct: there is no information contained about the future in X (over $[0,T]$). In other words, the authors “extrapolate”, while their information theoretical measures only contain information that allows them to “interpolate” (within $[0,T]$). Furthermore, a clear and precise definition of “predictability” and “reconstructability” is missing and crucial here!

Is the objective of this manuscript to introduce the measures or is there some insight from their analysis. Either way, the objective of the manuscript and the interpretation of the measures should be clearly stated in the introduction.

A reader expects that the authors can “predict” and “reconstruct”, given the process X (over some time interval $[0,T]$).

A real step forward would be that

(a) a given system is both predicted and reconstructed (to some degree) and the accuracy of the prediction and the reconstructed graph is evaluated precisely based on a ground truth (i.e. simulate the outcome of a process on a certain graph over a time-interval $[0,T]$, where both process X and graph G are precisely known)

(b) the mutual information is computed for these specific cases of X and G and compared with the accuracy in (a).

Such a comparison would make the framework valuable. Although the article is well-written, I do not think that there is any substantial scientific value nor insight, apart from a suggestive one.

Major:

- Fig. 3: the size of graph is $N=5$ nodes and $E=5$ edges, which is really small. What is the reason for such small graphs? Are such graphs representable for real-world situations? If the aim is to consider all possible graphs with $N=5$ and $E=5$, then it should be clearly mentioned.

- Eq. (5) may need a little more comments. Is this why the authors constrain to 'random graphs'? This point of view needs more thought, because not all graphs with E edges and N nodes are uniform with respect to the process X . About 20 years of Network Science have clearly pointed out that not all graphs are uniform. For example, a process on a regular lattice is different (with the same nodes and edges) from a small-world network with a sufficient number of rewired links. Only a few of such random links make a serious difference in the diameter, clustering and other graph properties that strongly will impact the process. There are much more examples. Hence, the uniformity assumption over all graph is quite problematic. If that assumption is erroneous, then the author's probabilistic framework loses ground.

- If the graph does not change during the process, many examples can be given where prediction of the process X is accurate, but the reconstruction of the graph is hardly possible. Thus, the entire "duality" here is likely not correct.

Minor:

- Eq. (4) is not needed in the main text (overly complicated), while (3) is, indeed, the joint prob. of the vector X in any MC. The comment is general: I would place Theorem I in the main text and place details to the SI.

- Page 2, Fig. 1: it seems that $H(G) \gg H(X)$ and $H(G) \ll H(X)$ should be reversed.

- Page 6, end of subsection II.B ("We address this problem in the next section"): the computational aspects are discussed in Section IV, not in Section III.

- Page 6: it seems that there is a wrong reference to the figure: it should be Fig.4 instead of Fig. 5;

- Page 7, Section III, lower-case letter "a" in the sentence "The relationship between X and G could also go the other way around: A system in which X generates ...";

- Page 12, Graph enumeration approach subsection: I did not get the meaning of the graph-states pairs.

- The definition of X^* is not provided.

- Should Sections III (Discussion) and IV (Materials and Methods) be reversed? The authors always reference Section IV, so you should always get to the end of the paper.

Reviewer #2 (Remarks to the Author):

The paper "Duality between predictability and reconstructability in complex systems" focuses on an interesting problem in complex systems: given a set of timeseries of a dynamical process defined on

nodes, to what degree it is possible to reconstruct the underlying network of interactions among the nodes AND at the same time predict the future evolution of the system?

I find the idea of the paper extremely interesting and elegant (although the results are a bit depressing from the point of view of a practitioner).

The manuscript is very well structured (modulo one point raised below) and the material well explained.

The methodological contribution to the estimation of the quantities described is also very important.

Overall, I think the paper is a great contribution.

However, I have a few issues that I think should be addressed (below), before I can make a recommendation for publication in Nature Communications.

Major issues:

- can you show an explicit example of a reconstruction problem performed with more standard or existing techniques that displays the predicted decay in predictability versus predictability with larger amounts of data?

For example, if you were to simulate the same SIS on a network, reconstruct both network and dynamics using a stochastic block model approach (e.g. Peixoto, Tiago P. "Network reconstruction and community detection from dynamics." *Physical review letters* 123.12 (2019): 128301.); and then use the process parameters to predict, would the duality still be valid?

The reason for the question is that the example (Fig.2) reported in the paper focus on either reconstructing the structure or predicting the dynamics.

However, in some cases it is possible to infer parameters of both structural and dynamical models at the same time, and I wonder whether this is something that escapes the current formalization. Or if instead it can be recast in the same language, and the results remain effectively the same.

- is it possible to link these predictability measure to one or more existing ones (e.g. Prasse, Bastian, and Piet Van Mieghem. "Predicting network dynamics without requiring the knowledge of the interaction graph." *Proceedings of the National Academy of Sciences* 119.44 (2022): e2205517119., or Prasse, Bastian, and Piet Van Mieghem. "Predicting network dynamics without requiring the knowledge of the interaction graph." *Proceedings of the National Academy of Sciences* 119.44 (2022): e2205517119.)?

- I would move the whole section on the effects of conditioning on the past to the main text. I do not remember particular limits on size in Nat Comms, and I think it would paint a much more complete picture if it is right after the main results on T-duality, rather than somehow hidden in the appendices.

- on the subject of past information, I like how the problem was set up and explained. I have a doubt however: X and Y in this case are two parts of the same timeseries of the original (order 1) Markov chain, right?

If so, when comparing the various cases of τ vs T, we really are only considering the past versus the future in terms of amount of timepoints, not in terms of using a longer memory (higher-order) in the markov chain?

- is there a relation between the reconstructability for $\tau=T-5$ of the various processes and the reconstructability of the system for $\tau=1$ at time step ~ 5 ?

- How do the results change if the dynamical systems does not have a stationary state?

The results in the paper refer to a situation in which there is always a positive entropy rate and the states considered are from systems like SIS process above threshold, Glauber or Cowan dynamics, that have no absorbing states in the chosen regimes. Moreover, the authors briefly state that there is no information below the critical point.

However, what would happen if instead one considers an SIS under the threshold (or equivalently an SIR model), but with multiple iterations of the process (e.g. restarting everytime the epidemic disappears)?

It might appear a moot point, but many neurophysiological experiments effectively fall in this case (multiple trials of the same stimulus above a non-informative resting state).

Does the duality still hold in this type of processes?

In this case it seems to me that, for each iteration, $H(X)$ might not explode (as opposed to the stationary case), and thus it might not necessarily kill $U(X|G)$.

- Assuming that everything works, the title is a little misleading.

While I appreciate the design of the paper and the overall results, I feel that the title is a little too generic.

For example, the proposed approach computationally can only work for systems with discrete states (binary or not), and it is unclear whether it could be directly generalized to the case of continuous dynamical systems (e.g. oscillators) and observations (e.g. ecological or neuroimaging data), which constitute a majority of real-world applications.

Minor:

= there is a broken sentence in the caption of Figure 6 ("..time steps T, and .").

Response to Reviewer 1

Comment 1.1

The authors have proposed an information theoretic approach to the dependence between the process X on a graph and the structure of that graph G . The manuscript is well written and the story seems compelling at first glance!

Reply 1.1

We thank the Reviewer for acknowledging the quality of our work.

Comment 1.2

Nevertheless, there are serious concerns. The authors have interpreted the mutual information $I(X;G)$ in Eq. 1 (1a and 1b) in terms of the “predictability” and “reconstructability”. I think that the authors overaccentuate the meaning of “predictability” and “reconstructability”: how does mutual entropy of X and G translates to “prediction” and “reconstructability”; mathematically, there is only a probabilistic conditional dependence between G and X and vice versa. Instead of predictability, we may talk e. g. about “sensitivity” of X upon G (and vice versa), which is more correct: [...]

Furthermore, a clear and precise definition of “predictability” and “reconstructability” is missing and crucial here!

Reply 1.2

Even though it is true that G and X are related through a probabilistic conditional dependence, it does allow to define two distinct measures, namely the predictability and reconstructability. In what follows, we hope to convince the Reviewer that it is a legitimate terminology. In the main paper, we exposed a detailed description of our measures in Section II.A, with intuitive examples, that clarified this choice of terms. We even showed in this section that the mutual information, the foundation of our work, is related to the empirical performance accuracy of reconstruction algorithms and—now, in the new Fig. 2—of prediction algorithms as well. However, we understand that further clarifications are needed since both reconstructability and predictability are loaded terms, especially in the Complex Systems community, that we must use carefully.

Let us first consider the case of reconstructability, as we believe it is a less frequently used term. In our work, we define reconstructability as the fraction of the information about a graph G that can be recovered from observing a process X . Reconstructability is measured by $U(G|X) = \frac{I(X;G)}{H(G)}$ (now Eq. (2a) of the main paper). Note that the conditional probability $P(G|X)$, directly used to compute $I(X;G)$, has been used for Bayesian network reconstruction [1–4]. Intuitively, the mutual information quantifies how different the reconstruction—posterior—model $P(G|X)$ is when compared to the prior model $P(G)$. The mutual information $I(X;G)$ is the average log-likelihood ratio of the two, to be more precise. As a result, and including the previously mentioned evidence of new Fig. 2, we believe it is reasonable to call $U(G|X)$ the *reconstructability*.

The predictability is perhaps the more complex case of the two, since it has been used in various fields and therefore has carried diverse technical definitions, namely those based on information theory [5–11]. In this sense, there is no unique notion of predictability, but rather a large spectrum of predictability measures that provide different insights on a system. For example, Ref. [8, Equation 18] presents measures of predictability expressed as the mutual information $I(X_1; X_T)$ between the initial conditions of a process X , denoted X_1 , and the state of the system at a later time T , denoted X_T . Other works like Ref. [5, Equation 9] and [7, Equation 9] also considered a more general formulation of mutual information as a measure of predictability, such as $I(\theta; X_T)$, where θ represents some general parameter (typically a vector) related to the initial conditions. In our context, we define predictability as the fraction of information about the process X that can be extracted from the graph G , and it is measured by $U(X|G) = \frac{I(X;G)}{H(X)}$ (Eq. (2b) of the main paper). Our formulation of predictability focuses on the contribution of the graph to our ability

to predict the process, rather than the initial conditions. However, one can argue that G is in fact a parameter related to the initial conditions of the systems. This makes our predictability compatible with the one proposed in Ref. [5]. The Reviewer has mentioned the term "sensitivity", and we thank them for it as it is a literature that we had overlooked. Indeed, from Refs. [12, 13], we understand that our predictability measure is quite close to some parameter sensitivity analysis measures that quantify the influence of a function's parameters over its output. Whether it is a "more correct" term than predictability is, in our opinion, debatable. Predictability in the conventional sense is arguably a sensitivity measure in itself. From our perspective, the difference between sensitivity and predictability comes from the temporal nature of X and the ordering between G and X (it is G that generates X , not the other way around). Therefore, we believe it makes our use of predictability a bit more precise than sensitivity.

Action taken 1.2

To further clarify our nomenclature, we consolidated our definitions of predictability and reconstructability in Section II.A of the paper. Specifically, we now mention the term "sensitivity" in this section to describe our predictability. Moreover, we include new Section II.C which presents simple examples where our measures can be analytically computed and are easy to interpret. Hopefully, it will be easier for the reader to understand our choice of terminology through these new examples.

Comment 1.3

[...] there is no information contained about the future in X (over $[0, T]$). In other words, the authors "extrapolate", while their information theoretical measures only contain information that allows them to "interpolate" (within $[0, T]$).

Reply 1.3

The Reviewer raises an interesting question on whether our measures are interpolating or extrapolating in the interval $[1, T]$ [14], which we had not considered up until now. It is our understanding that this comment only applies to the predictability measure, as the notion of "predicting within $[1, T]$ " does not make sense in the context of reconstructing the graph G .

It is true that $U(X|G)$ can only inform us about the process X within the interval $[1, T]$, naturally because X is defined only within that interval. Nevertheless, it is not correct to say that our measure interpolates. On the contrary, it extrapolates. To be clear, what we mean by interpolation implies to fill the gaps between a set of non-consecutive known states: for instance predicting the states within $(1, T)$ given X_1 and X_T . Rather, extrapolation corresponds to predicting outside the interval of known states, for instance within $(1, T]$ given only the initial conditions X_1 . The partial measures, previously presented in Section IV.C (now Section II.B), that leveraged the conditional mutual information $I(X_{\text{future}}; G|X_{\text{past}})$, generalize those defined from $I(X; G)$ by including the past. In this case, we explicitly know which parts of the time series are included in the prediction in-sample, which one is predicted out-of-sample. By relating $I(X; G)$ and $I(X_{\text{future}}; G|X_{\text{past}})$, it becomes clear that our predictability measures really are extrapolating.

Action taken 1.3

We mention the extrapolating nature of the predictability in Section II.A.

Comment 1.4

Is the objective of this manuscript to introduce the measures or is there some insight from their analysis. Either way, the objective of the manuscript and the interpretation of the measures should be clearly stated in the introduction.

Reply 1.4

To answer the question plainly, we do both: we introduce measures allowing us to gain insights about the structure-function relationship in complex systems. As it is mentioned in the last paragraph of Section I, our work allowed us to unify predictability and reconstructability in a single framework which, to our knowledge, is unheard of. Furthermore,

we identified a new phenomenon, namely the predictability-reconstructability duality, which may have practical implications in neuroscience. There, network reconstruction and prediction are deeply intertwined: reconstruction is employed to infer a structure from time series data, producing what is known as a functional connectome, which is often used to assess the evolution of brain's states, thus aligning with our definition of prediction. Also, the fact that criticality might play an important role for the predictability-reconstructability duality is also a contribution of our framework.

Comment 1.5

A reader expects that the authors can “predict” and “reconstruct”, given the process X (over some time interval $[0, T]$). A real step forward would be that

- (a) a given system is both predicted and reconstructed (to some degree) and the accuracy of the prediction and the reconstructed graph is evaluated precisely based on a ground truth (i.e. simulate the outcome of a process on a certain graph over a time-interval $[0, T]$, where both process X and graph G are precisely known);*
- (b) the mutual information is computed for these specific cases of X and G and compared with the accuracy in (a). Such a comparison would make the framework valuable. Such a comparison would make the framework valuable. Although the article is well-written, I do not think that there is any substantial scientific value nor insight, apart from a suggestive one.*

Reply 1.5

From our understanding, the Reviewer is asking how the mutual information $I(G; X)$ compares with the performance accuracy while reconstructing G from X or predicting X from G . This was the intention behind Fig. 2 of the paper: comparing the performance accuracy of standard algorithms with $I(X; G)$, for both network reconstruction and time series prediction. It turns out that doing this for network reconstruction was more straightforward than for the time series prediction, which is why we only included the former at first. Fortunately, this can be done by using a very similar analysis to that presented in Ref. [15], where the true model is compared with another model via a mean absolute error. In their case, the true model is a deterministic dynamics that generates time series on a graph G , and the compared model is the same dynamics but where the structure is given by \hat{G} , a graph reconstructed from time series generated with G . The aim of their paper was to show that prediction can sometimes be achieved with great accuracy (low absolute error) without the knowledge of the graph.

Our case is quite similar to theirs, except we quantify the prediction accuracy with any graphs, not just reconstructed ones. Hence, our model of comparison is a graph-independent model, i.e. a model that predicts X given G , that mimics the marginal process occurring with probability $P(X)$. In general, the marginal process is non-Markovian which can be difficult to work with, but it can be closely approximated by Markovian heuristics such as logistic regressions and multilayer perceptrons. These heuristics are trained to map the state X_{t-1} of the system at time $t - 1$, a vector of zeros and ones of size N , to its future state X_t , while taking into account the random nature of the process. As a result, these models output a set of transition probabilities that can be directly compared to those used for computing $P(X|G)$.

The mean absolute error between the conditional and marginal models probabilities is strongly correlated with $I(X; G)$, as shown by the new Fig. 2(a). On that one hand, a small error implies that the marginal model is close to the true graph-dependent model, which in turn indicates that G might not be useful for predicting X . In this case, we also observe a small mutual information. On the other hand, a large error implies that the two models are quite different from one another, indicating that G might have a strong influence over X , i.e. a large mutual information.

Action taken 1.5

We now include a comparison between the time series prediction accuracy of different algorithms in Fig. 2(a), and

describe this experiment in Section II.A.

Comment 1.6

Fig. 3: the size of graph is $N = 5$ nodes and $E = 5$ edges, which is really small. What is the reason for such small graphs? Are such graphs representable for real-world situations? If the aim is to consider all possible graphs with $N = 5$ and $E = 5$, then it should be clearly mentioned.

Reply 1.6

We acknowledge that $N = 5$ corresponds to very small graphs. The point of this section is to illustrate $U(G|X)$ and $U(X|G)$ using unbiased estimators while showing the T -duality, which could be quite computationally expensive to illustrate on larger graphs. As mentioned in now Section IV.E, this unbiased estimator necessitates graph enumeration, for which with larger N the evaluation is virtually impossible to complete.

However, our analysis is not restricted to small graphs, since we used biased estimators of $I(X;G)$ that bounds it from below and above. These estimators can efficiently be evaluated on larger graphs, as shown in now Fig. 7 of the main paper and in supplementary Fig. 3. This is in fact a key contribution of the paper to be able to apply our methodology to larger graphs, which is highlighted by Referee 2, “The methodological contribution to the estimation of the quantities described is also very important.”.

Action taken 1.6

We now state more clearly that the use of small graphs is to allow exact calculations to illustrate our measures, since their computation are very computationally expensive. We then readily explain that the bounds introduced afterwards allows calculations for larger graphs to avoid confusions.

Comment 1.7

Eq. (5) may need a little more comments. Is this why the authors constrain to ‘random graphs’? This point of view needs more thought, because not all graphs with E edges and N nodes are uniform with respect to the process X . About 20 years of Network Science have clearly pointed out that not all graphs are uniform. For example, a process on a regular lattice is different (with the same nodes and edges) from a small-world network with a sufficient number of rewired links. Only a few of such random links make a serious difference in the diameter, clustering and other graph properties that strongly will impact the process. There are much more examples. Hence, the uniformity assumption over all graph is quite problematic. If that assumption is erroneous, then the author’s probabilistic framework loses ground.

Reply 1.7

By ‘random graphs’, we meant *any ensemble of graphs*, or random network models. in the general sense [16]. These are governed by a general probability distribution $P(G)$, not just uniformly distributed random graphs. Thus, the random variable G can be as complicated as needed. The Erdős-Rényi model, sometimes called the *random graph model*, was used throughout the paper (now Figs. 2, 5 and 8) for its simplicity only—not by the limitation of our framework. We are aware that many sophisticated random graph models exist to accommodate different modeling assumptions. One of them is the configuration model (Eq. 7 of the main paper), which does display a non-uniform distribution over all graphs. In fact, we used this model for Fig. 4—now Fig. 7—, something the Reviewer may have overlooked. However, we fixed the degree distribution to be a geometric distribution, which might leave the reader’s thirst for realism unquenched. This is why we now also consider the degree distributions of real networks in now Fig. 7 to further illustrate our point.

Note that it would be possible to go beyond the Erdős-Rényi and the configuration models, but we think going in that direction is beyond the scope of current paper. For instance, we could consider models as sophisticated as

the stochastic block model (SBM) used in Ref. [2], where both the graph and the community structure are inferred simultaneously. However, while our theoretical framework could handle the SBM, in practice the numerical procedure would be substantially more complex to perform. We feel that including such model would complicate unnecessarily the manuscript without bringing much to the history we tell regarding the duality. For these reasons, we decided to leave it to future works (this project is in fact ongoing).

Action taken 1.7

In light of the Reviewer’s comment, we also include the degree distribution of real networks in now Fig. 7 (bottom row) to display the relationship between duality and criticality.

Comment 1.8

If the graph does not change during the process, many examples can be given where prediction of the process X is accurate, but the reconstruction of the graph is hardly possible. Thus, the entire “duality” here is likely not correct.

Reply 1.8

We suppose that the Reviewer’s comment is likely related to Ref. [15], among others, which showed that the time series of some deterministic dynamics on a graph G can be predicted from the dynamics on a reconstructed graph \hat{G} that is significantly different from G . Yet, we believe that the examples mentioned by the Referee and the observations of Ref. [15] remain to be understood on a more detailed basis, where the predictability and the reconstructability are *both quantified and compared*. This is the gap that we aimed to fill. From our knowledge, we established the very first steps toward a general information-theoretic framework where both predictability and reconstructability are quantified and treated *together*. We found, first, that there is a relationship between the two and, second, that a high reconstructability does not necessarily imply a high predictability and vice versa. It is in this context that we observed the duality phenomenon between reconstructability and predictability. In this sense, our work offers complementary theoretical insights on the observations of Ref. [15] while providing a detailed analysis of how predictability and reconstructability change with respect to different parameters (if they change similarly or in opposite directions).

On another front, it is important to mention that the intention of our paper was never to claim the universality of the duality between predictability and reconstructability—except for the T -duality—, but simply to state its existence. On many occasions, we illustrate scenarios where the duality exists and strongly emphasize on these examples to better characterize the phenomenon mathematically and numerically. However, we also show examples where the duality does not appear (Fig. 4—now new Fig. 7—outside of the shaded areas, Fig. 6 (f) and (i)—now new supplementary Fig. 2). Hence, the fact that the duality might not appear in some scenario does not contradict the narrative of the paper, and certainly does not contradict the findings of Ref. [15].

Action taken 1.8

We now investigate the case of deterministic dynamics in Supplementary Note IV, and elaborate a connection between our work and Ref. [15].

Comment 1.9

Eq. (4) is not needed in the main text (overly complicated), while (3) is, indeed, the joint prob. of the vector X in any MC. The comment is general: I would place Theorem I in the main text and place details to the SI.

Reply 1.9

We agree with the Referee that rearranging the equations to put more emphasis on the salient results is more appropriate.

Action taken 1.9

We placed Eq. (4) in the Material and Methods section (now Eqs. (8) and (9)), moved Eq. (3) in the text and moved the formal statement of Theorem I in the main text.

Comment 1.10

Page 2, Fig. 1: it seems that $H(G) \gg H(X)$ and $H(G) \ll H(X)$ should be reversed.

Action taken 1.10

This typo has been corrected.

Comment 1.11

Page 6, end of subsection II.B ("We address this problem in the next section"): the computational aspects are discussed in Section IV, not in Section III.

Action taken 1.11

This has been corrected.

Comment 1.12

Page 6: it seems that there is a wrong reference to the figure: it should be Fig.4 instead of Fig. 5;

Action taken 1.12

This has been corrected.

Comment 1.13

Page 7, Section III, lower-case letter "a" in the sentence "The relationship between X and G could also go the other way around: A system in which X generates ... ";

Action taken 1.13

This has been corrected.

Comment 1.14

Page 12, Graph enumeration approach subsection: I did not get the meaning of the graph-states pairs. The definition of X^ is not provided.*

Reply 1.14

By graph-state pairs, we mean that for every time series generated from X , denoted as the state X^* , we used a graph G^* sampled from G . In other words, G^* and X^* are realizations of G and X , respectively, and we denote (G^*, X^*) as a graph-state pair, implying that G^* was used to generate X^* .

Action taken 1.14

To avoid confusion, we removed this nomenclature of the main text. Additionally, we replaced the notation for realization of random variables (for instance, G^* and X^* being realizations of G and X , respectively) to a more standard notation, where realizations are denoted by lower-cased variables.

Comment 1.15

Should Sections III (Discussion) and IV (Materials and Methods) be reversed? The authors always reference Section IV, so you should always get to the end of the paper.

Reply 1.15

We agree with the Reviewer that this is not the most conventional way to organize our manuscript, as it forces the reader to go back and forth between the main text and the Material and Methods section. It is however, the format adopted by Nature Communications.

Response to Reviewer 2

Comment 2.1

The paper "Duality between predictability and reconstructability in complex systems" focuses on an interesting problem in complex systems: given a set of time series of a dynamical process defined on nodes, to what degree it is possible to reconstruct the underlying network of interactions among the nodes AND at the same time predict the future evolution of the system?

I find the idea of the paper extremely interesting and elegant (although the results are a bit depressing from the point of view of a practitioner). The manuscript is very well structured (modulo one point raised below) and the material well explained. The methodological contribution to the estimation of the quantities described is also very important. Overall, I think the paper is a great contribution. However, I have a few issues that I think should be addressed (below), before I can make a recommendation for publication in Nature Communications.

Reply 2.1

We thank the Reviewer for such an accurate description of our work. We are also delighted that they find our contribution "extremely interesting and elegant" and "very important", and that our manuscript is overall "well structured" and "well explained". We address each of their comments below, and hope to convince them to recommend our revised manuscript for publication in Nature Communications.

Comment 2.2

*Can you show an explicit example of a reconstruction problem performed with more standard or existing techniques that displays the predicted decay in predictability versus [reconstructability] with larger amounts of data? For example, if you were to simulate the same SIS on a network, reconstruct both network and dynamics using a stochastic block model approach (e.g. Peixoto, Tiago P. "Network reconstruction and community detection from dynamics." *Physical review letters* 123.12 (2019): 128301.); and then use the process parameters to predict, would the duality still be valid? The reason for the question is that the example (Fig.2) reported in the paper focus on either reconstructing the structure or predicting the dynamics. However, in some cases it is possible to infer parameters of both structural and dynamical models at the same time, and I wonder whether this is something that escapes the current formalization. Or if instead it can be recast in the same language, and the results remain effectively the same.*

Reply 2.2

There are two things to address here. First, the Reviewer asked whether it would be possible to show the predictability-reconstructability duality using standard techniques (hopefully, similar to those used in new Fig. 2 (a) and (b)), for instance, as a function of the number of time steps T —or, the amount of data, as they put it. There are several challenges to solve before we can do this. Recall that the marginal model governed by $P(X)$, which the heuristic techniques are trying to replicate, is generally non-Markovian; something these models are not designed to capture. As a result, the error related to the Markov nature of the heuristics accumulate when T increases and the absolute error therefore cannot scale similarly to $I(X;G)$ in this specific case. This was not a problem for new Fig. 2(a), since T remained fixed, and the error related to the Markov property remained of the same magnitude in each case. Additionally, we used the mean absolute error to quantify predictability, which does not scale with T like $I(X;G)$ does. To illustrate the duality with T using heuristics, we would have to find an error measure that scales with T similarly to $I(X;G)$, and find heuristics that incorporates the non-Markovian nature of $P(X)$. As of today, we did not find effective ways to do it.

However, using the techniques now in the paper, we can compare the duality predicted by our framework with the dual behavior of prediction and reconstruction accuracies (see new Supplementary Fig. 1). The only difference is

that, instead of using the mean absolute error directly as a proxy of $I(X; G)$, we normalize it with the error of the graph-independent model with the binary time series to mimic $U(X|G)$. Again, our measures are well correlated with the prediction and reconstruction accuracies, which further brings more evidence to the existence of the duality.

Second, the Reviewer asks if the duality would remain valid if, instead of using the parameters of the model, we used parameters inferred from time series alongside with a reconstructed graph (like in Ref. [2]). This is an interesting question, as it begs the question if our framework could actually be used in a real Bayesian inference setting: if we are given some observations X and we reconstruct both G and the parameters of X , can we still assess the existence of the duality? The short answer is no, because the duality in our work is defined in terms of variations of $U(G|X)$ and $U(X|G)$. If no parameter can be varied, like in Bayesian inference where the observations X are constant, the duality cannot be investigated. However, this does not imply that (i) our measures cannot be computed or that (ii) they are not meaningful in a real Bayesian setting.

Furthermore, it begs for a generalized version of our work, involving additional parameters for X and for G . Suppose G is parametrized by θ and X by ϕ , such that their joint probability is factored as follows:

$$P(\theta, G, \phi, X) = P(\theta)P(\phi)P(G|\theta)P(X|G, \phi). \quad (1)$$

In other words, the variables (θ, ϕ, G, X) form the Bayesian network $\theta \rightarrow G \rightarrow X \leftarrow \phi$. For ϕ , an obvious choice would be the parameters of the process itself, like the coupling parameter J in the Glauber dynamics. Likewise, additional graph parameters θ could be, for instance, the partitioning of the nodes in the stochastic block model [2]. Then, the direct generalization of $I(X; G)$ would be $I(\phi, X; \theta, G)$, which puts all the dynamic information on one side, and the structural information on the other side. Remarkably, it is easy to show that $I(\phi, X; \theta, G) = I(X; G)$, because the only variables that are not conditionally independent are X and G :

$$\begin{aligned} I(\phi, X; \theta, G) &= H(\theta, G) - H(\theta, G|X, \phi) \\ &= H(\theta, G) - H(\theta, G|X) \quad [\text{by the conditional independence of } (\theta, G) \text{ and } \phi] \\ &= H(G) + H(\theta|G) - H(\theta|G, X) - H(G|X) \quad [\text{by the decomposition of entropies}] \\ &= H(G) + H(\theta|G) - H(\theta|G) - H(G|X) \quad [\text{by the conditional independence of } \theta \text{ and } X] \\ &= I(G; X) \end{aligned}$$

It is also possible to assess the mutual information of other relationships in this generalized framework. For instance, that $I(\phi; G) = 0$, meaning that nothing can be reconstructed from G if ϕ is only known. Also, that $I(X; \theta) = H(X|G) - H(X|\theta) + I(X; G) \leq I(X; G)$, which stipulates θ can be inferred from X and that this information is contained in $I(G; X)$.

Hence, to answer the second question, our framework is, in principle, generalizable to include additional parameters as well. In turn, we expect that a duality could still exist in that new setting. However, we think that providing a more concrete example of this, in terms of numerical simulations, would be out of the scope of the paper, for reasons mentioned in Reply 1.7.

Action taken 2.2

We include new Supplementary Note I—along with supplementary Fig. 1—that discuss the manifestation of the duality through prediction and reconstruction algorithms.

Comment 2.3

Is it possible to link these predictability measure to one or more existing ones (e.g. Prasse, Bastian, and Piet Van Mieghem. "Predicting network dynamics without requiring the knowledge of the interaction graph." Proceedings of the National Academy of Sciences 119.44 (2022): e2205517119.)?

Reply 2.3

This is an excellent question, and we thank the Reviewer for asking it. We believe that there is definitely a connection between our work and that of Prasse and Van Mieghem, as we also discussed in Reply 1.5. In their paper, they develop a procedure to reconstruct a graph assuming that the process is deterministic and governed by a set of differential equations. They find that, in their setting, many different graphs that satisfy a specific condition (Eq. [5] of their paper) can produce the desired outcome. They quantify predictability with the mean absolute error between the surrogate trajectories and the true ones, and reconstructability with the AUC of the reconstructed graph. Among other things, they show that high prediction accuracy can be achieved even when the graphs are weakly reconstructible (AUC close to 0.5).

The main differences between our work and theirs is (i) we used stochastic processes, and (ii) we use information theory to quantify predictability and reconstructability. Consequently, their conclusions are complementary and aligned to ours, but in a different setting. Also, our results are independent of the choice of inference algorithms, contrary to theirs which rely on a LASSO algorithm.

In addition to the ones of Prasse and Van Mieghem, there are other measures that closely resemble $U(X|G)$ used to measure predictability. The ones mentioned in the paper from Refs. [5–11] measures the sensitivity of the process X to its initial conditions, and many of them use different mutual informations directly as a measure of predictability (see Reply 1.2). Generally, our predictability measure can be interpreted in the context of parameter sensitivity analysis, as Reviewer 1 thankfully pointed out. Some works even quantified sensibility with mutual information and uncertainty coefficients [12, 13].

Action taken 2.3

As mentioned in Reply 1.5, we now use a similar analysis to the one presented in Ref. [15] to also quantify predictability (new Fig. 2). We also use our framework to investigate the predictability and reconstructability of deterministic dynamics like in Ref. [15] (see Supplementary Note IV), where we reach similar conclusions. We also mention the works of Refs. [12, 13] at the end of Section II.A where the uncertainty coefficients are presented.

Comment 2.4

I would move the whole section on the effects of conditioning on the past to the main text. I do not remember particular limits on size in Nat. Comm., and I think it would paint a much more complete picture if it is right after the main results on T-duality, rather than somehow hidden in the appendices.

Action taken 2.4

This is a great idea. This section has now been moved to the main text in the new Section II.B. The results in the former Fig. 5 have been moved to Section II.D in a new figure (also numbered 5). We also included a more detailed analysis of the past-dependent measure in Supplementary Note VIII. Furthermore, we change the notation from $Z \rightarrow X$, $X \rightarrow X_{\text{past}}$ and $Y \rightarrow X_{\text{future}}$ so that it remains more consistent with the rest of the paper.

Comment 2.5

On the subject of past information, I like how the problem was set up and explained. I have a doubt however: X and Y in this case are two parts of the same time series of the original (order 1) Markov chain, right? If so, when comparing the various cases of τ vs T , we really are only considering the past versus the future in terms of amount of time points, not in terms of using a longer memory (higher-order) in the Markov chain?

Reply 2.5

The Reviewer is right, this is in fact what we wanted to illustrate in former Fig. 5 (a). Although our framework is applicable to non-Markovian dynamics, we choose to limit our work to Markovian processes for simplicity, which

already offered various technical challenges. It would indeed be interesting to investigate non-Markovian processes with long memories, but we believe it falls outside the scope of the paper.

Comment 2.6

Is there a relation between the reconstructability for $\tau = T - 5$ of the various processes and the reconstructability of the system for $\tau = 1$ at time step ~ 5 ?

Reply 2.6

This is a good question. There is definitely a similarity between the two measures: When $\tau = T - 5$, the information we are given is a process of 5 time steps, and when $\tau = 1$ and $T = 6$, the given information is also a process of 5 time steps. In both cases, $U(X_{\text{future}}|G; X_{\text{past}})$ as given by—now, new—Eq. (4a) measures the fraction of information about G that is recoverable from a process of length 5, Y . The difference is with respect to the very information we are reconstructing. When $\tau = 1$ and $T = 6$, it is the whole graph G that we want to reconstruct, whereas it is a partially reconstructed graph that we want to reconstruction when $\tau = T - 5$. Indeed, when using $U(G|X_{\text{future}}; X_{\text{past}})$, we assume that the information contained in X has already been extracted and we are interested in the reconstruction of the remaining information of G . Hence, while they are very analogous, they actually mean different things.

Action taken 2.6

We added a comment in the new Section II.B regarding the past-dependent measures that explains the similarity and differences between the two scenarios, but in a more general setting.

Comment 2.7

How do the results change if the dynamical systems does not have a stationary state? The results in the paper refer to a situation in which there is always a positive entropy rate and the states considered are from systems like SIS process above threshold, Glauber or Cowan dynamics, that have no absorbing states in the chosen regimes. Moreover, the authors briefly state that there is no information below the critical point. However, what would happen if instead one considers an SIS under the threshold (or equivalently an SIR model), but with multiple iterations of the process (e.g. restarting every time the epidemic disappears)? It might appear a moot point, but many neurophysiological experiments effectively fall in this case (multiple trials of the same stimulus above a non-informative resting state). Does the duality still hold in this type of processes? In this case it seems to me that, for each iteration, $H(X)$ might not explode (as opposed to the stationary case), and thus it might not necessarily kill $U(X|G)$.

Reply 2.7

This is a very good point that we have not discussed in the paper. For non-stationary processes like the Reviewer is intending, the entropy rate might not be well-defined. However, while the assumption that the entropy rate is well defined and positive is needed in order to complete the proof of Theorem 1 (for which the T -duality holds for all $T \geq \tau$), it does not exclude the possibility of having regions of T -duality for non-stationary processes. Apart from a T -duality, they might also exhibit other θ -dualities. It would be interesting to examine this in the future.

Action taken 2.7

We discuss the implication of Theorem 1 for non-stationary processes in the main text, after its formulation in new Section II-D.

Comment 2.8

Assuming that everything works, the title is a little misleading. While I appreciate the design of the paper and the overall results, I feel that the title is a little too generic. For example, the proposed approach computationally can only work for systems with discrete states (binary or not), and it is unclear whether it could be directly generalized to the

case of continuous dynamical systems (e.g. oscillators) and observations (e.g. ecological or neuroimaging data), which constitute a majority of real-world applications.

Reply 2.8

We agree with the Reviewer that our approach is only used on discrete states systems, in the current paper. However, we must stress that in the general formulation of our framework, the discrete state assumption is not needed. That is, in principle, it is possible to evaluate analogs to $I(X;G)$, $U(X|G)$ and $U(G|X)$ in continuous state systems that involve the probability density functions $\rho(X)$ and $\rho(X|G)$ instead of probability mass functions $P(X)$ and $P(X|G)$, and where the sum over X to recover the normalization of $\rho(X)$ and $\rho(X|G)$ is replaced by an integral. The mathematical analyses presented in Sections IV.A and IV.B (now Sections IV.C and IV.D remain valid even for continuous state X , as long as X has the Markov property.

That being said, in practice, the evaluation of $I(X;G)$, $U(X|G)$ and $U(G|X)$ is technically more involved for continuous state systems than for binary state systems, which is why we focus our analysis on the latter. What would be needed in order to generalize our numerical analysis to continuous state would be any class of systems where the density function $\rho(X|G)$ can be efficiently evaluated. Then, the numerical procedures presented in now Section IV.E could be used directly.

Nevertheless, for the reasons mentioned above, we believe that the generality of the title is valid, and even critical for the message we want to send to the complex systems community. Which is why we propose to keep the title as it is, even though a clarification relating to continuous state systems is needed in the manuscript to justify our general title.

Action taken 2.8

We now mention the possibility to use our framework also for continuous-state systems at the beginning of Section II.A. As previously mentioned, we also include a calculation of $U(G|X)$ and $U(X|G)$ for deterministic dynamics with continuous states in Supplementary Note IV.

Comment 2.9

= *there is a broken sentence in the caption of Figure 6 ("..time steps T, and . ")*.

Action taken 2.9

The sentence has been corrected.

-
- [1] T. P. Peixoto, “Reconstructing networks with unknown and heterogeneous errors,” *Phys. Rev. X* **8**, 041011 (2018).
 - [2] T. P. Peixoto, “Network reconstruction and community detection from dynamics,” *Phys. Rev. Lett.* **123**, 128301 (2019).
 - [3] J.-G. Young, G. T. Cantwell, and M. E. J. Newman, “Bayesian inference of network structure from unreliable data,” *J. Complex Netw.* **8**, cnaa046 (2020).
 - [4] J.-G. Young, F. S. Valdovinos, and M. E. J. Newman, “Reconstruction of plant–pollinator networks from observational data,” *Nat. Commun.* **12**, 3911 (2021).
 - [5] T. DelSole and M. K. Tippett, “Predictability: Recent insights from information theory,” *Rev. Geophys.* **45**, RG4002 (2007).
 - [6] C. Song, Z. Qu, N. Blumm, and A.-L. Barabási, “Limits of predictability in human mobility,” *Science* **327**, 1018 (2010).
 - [7] D. Giannakis, A. J. Majda, and I. Horenko, “Information theory, model error, and predictive skill of stochastic models for complex nonlinear systems,” *Physica D* **241**, 1735–1752 (2012).
 - [8] R. Kleeman, “Information theory and dynamical system predictability,” *Entropy* **13**, 612 (2011).
 - [9] J. Garland, R. James, and E. Bradley, “Model-free quantification of time-series predictability,” *Phys. Rev. E* **90**, 052910 (2014).
 - [10] F. Pennekamp, A. C. Iles, J. Garland, G. Brennan, U. Brose, U. Gaedke, U. Jacob, P. Kratina, B. Matthews, S. Munch, M. Novak, G. M. Palamara, B. C. Rall, B. Rosenbaum, A. Tabi, C. Ward, R. Williams, H. Ye, and O. L. Petchey, “The intrinsic predictability of ecological time series and its potential to guide forecasting,” *Ecol. Monogr.* **89**, e01359 (2019).
 - [11] S. V. Scarpino and G. Petri, “On the predictability of infectious disease outbreaks,” *Nat. Commun.* **10**, 1 (2019).
 - [12] G. C. Critchfield, K. E. Willard, and D. P. Connelly, “Probabilistic sensitivity analysis methods for general decision models,” *Comput. Biomed. Res.* **19**, 254–265 (1986).
 - [13] N. Lüdtke, S. Panzeri, M. Brown, D. S. Broomhead, J. Knowles, M. A. Montemurro, and D. B. Kell, “Information-theoretic sensitivity analysis: a general method for credit assignment in complex networks,” *J. R. Soc. Interface* **5**, 223–235 (2008).
 - [14] It is our convention that we start at $t = 1$, thus $t = 0$ is not included in X .
 - [15] B. Prasse and P. Van Mieghem, “Predicting network dynamics without requiring the knowledge of the interaction graph,” *Proc. Natl. Acad. Sci. U.S.A.* **119**, e2205517119 (2022).
 - [16] B. Bollobas, *Random Graphs*, 2 ed. (Cambridge University Press, 2001).

REVIEWERS' COMMENTS

Reviewer #3 (Remarks to the Author):

I read with interest the manuscript by Murphy and collaborators dealing with the duality between predictability and reconstructability in networks. The paper went under a previous round of review, and it seems to me that the authors satisfactorily addressed many of the comments they received. I found the revised version of the manuscript well written and organized, and I believe that the theory developed in this work is a very important contribution in the area of complex systems research. I am therefore in favor of publication in the journal.

Although my recommendation is positive, I still have a few minor comments (mostly related to the presentation rather than to the actual content of the paper) that I hope the authors will address before publication:

1) Based on my understanding, the "predictability" metric $U(X|G)$ quantify the relative amount of additional information that is required to make a prediction about the dynamical state of the network even if the graph structure is known. As such, the metric equals zero even when the system's dynamical state is fully predictable, whereas equals one when the dynamical state is completely unpredictable. For example in Fig.5b, in the long-term regime of SIS dynamics all nodes are likely to be in the S state, thus the system's dynamical state is fully predictable. I find this definition a bit counterintuitive. Wouldn't it be more direct to define the predictability as $1 - U(X|G)$?

2) Just to iterate on point 1. I would define reconstructability as $1 - U(G|X)$. It's just more intuitive to me: high value of the reconstructability metric = high ability to reconstruct the graph, low value of the reconstructability metric = low ability to reconstruct the graph.

3) As I mentioned already, the manuscript is well written and organized. Also, I appreciate the fact that mathematical details are postponed in the appendix and SI. In spite of this, I still find paper excessively long and dense. Somehow, this makes it a little difficult to grasp the main message of the paper, which instead is very intuitive, elegant and important. To increase the future impact of the work and make it accessible to a larger/less specialist readership, I would try to make the paper more concise.

4) The authors may want to cite papers on predictability methods based on maximum entropy sampling
J. Mach. Learn. Res. 9, 235 (2008)

Phys. Rev. Lett. 120, 198301 (2018)

NCOMMS-23-10333

Duality between predictability and reconstructability in complex systems

Dated: March 5, 2024

Dear Editor and Referee,

We thank the editor for handling the unforeseen events related to the management of our paper and the Referee for providing a positive review of the article on such a short notice.

Following their comments, we have added a new relevant citation to the paper. The paper and the measures therein have remained the same as justified in our point-by-point reply to each comment.

Sincerely,

The authors.

Response to Reviewer 3

Comment 3.1

I read with interest the manuscript by Murphy and collaborators dealing with the duality between predictability and reconstructability in networks. The paper went under a previous round of review, and it seems to me that the authors satisfactorily addressed many of the comments they received. I found the revised version of the manuscript well written and organized, and I believe that the theory developed in this work is a very important contribution in the area of complex systems research. I am therefore in favor of publication in the journal.

Reply 3.1

We thank the reviewer for their positive comments about our manuscript, and support regarding the previous round of review.

Comment 3.2

Based on my understanding, the "predictability" metric $U(X|G)$ quantify the relative amount of additional information that is required to make a prediction about the dynamical state of the network even if the graph structure is known. As such, the metric equals zero even when the system's dynamical state is fully predictable, whereas equals one when the dynamical state is completely unpredictable. For example in Fig.5b, in the long-term regime of SIS dynamics all nodes are likely to be in the S state, thus the system's dynamical state is fully predictable. I find this definition a bit counterintuitive. Wouldn't it be more direct to define the predictability as $1 - U(X|G)$?

Reply 3.2

There seems to be some misunderstanding here. The measure $U(X|G)$, that we call predictability, quantifies the relative amount of information *acquired* about the dynamical state if the graph structure is known. To put it differently, it quantifies the relative *reduction* in uncertainty about X when G is known. The difference between our definition and the Reviewer's is whether the information is acquired or required. The measure the Reviewer is referring to is precisely the relative entropy of X given G , that is $\frac{H(X|G)}{H(X)}$. Therefore, from Eq. 2(a) of the manuscript,

$$U(X|G) = \frac{I(X;G)}{H(X)} = 1 - \frac{H(X|G)}{H(X)}, \quad (1)$$

we see that the Reviewer's suggestion is actually equivalent to our current definition of predictability, which reaches 1 when the system is fully predictable. Indeed, as stressed below Eq. 2(a) in the manuscript: *This maximum value is guaranteed when X is deterministic and there is only one initial condition (see Supplementary Note III)*. We invite the reviewer to carefully examine the section II.C. "Simple example", where the interpretation of reconstructability and predictability is firmly established.

The Reviewer presents an example where seemingly the predictability's behaviour is counterintuitive: In Fig. 5(b), for the SIS model in the long-term region, the predictability converges to zero even though, as mentioned by the Reviewer, the system likely fall into an absorbing state, where all nodes are in state S . This phenomenon is well explained by one of our main result in the paper, i.e., Theorem 1 of the manuscript. First, for the chosen values of infection rate $\lambda \in \{0.25, 0.5, 1\}$, the system likely does not fall into an absorbing state. Therefore, the state of the system constantly changes, which results in an ever increasing entropy of X with the length T of the process: The entropy rate of X is non-zero. In this regime, the information acquired from the graph G is limited—i.e., bounded by $H(G)$ —while the information needed to predict X is constantly increasing. This is why the predictability converges to zero as Theorem 1 predicts.

We thank the Referee for their interesting comment on the absorbing state, which is indeed a relevant and subtle

case. First note that Theorem 1 in the paper excludes this case, as the entropy rate is assumed to be non-zero. It is nevertheless partially covered in Supplementary Material, in the subsection “Markov process on a single node graph”. There, we can see that even when one graph leads to an absorbing state 0, it is still possible to have a predictability that converges to zero as $T \rightarrow \infty$. The reason is simple: one graph does not lead to an absorbing state in the dynamics, implying that the entropy $H(X)$ still increase as $T \rightarrow \infty$. We hope this helps in clarifying the matter.

Comment 3.3

Just to iterate on point 1. I would define reconstructability as $1 - U(G|X)$. It's just more intuitive to me: high value of the reconstructability metric = high ability to reconstruct the graph, low value of the reconstructability metric = low ability to reconstruct the graph.

Reply 3.3

Again, we refer the Reviewer to Eq. 2(b) and section II.C. of the manuscript, which show that the Reviewer's suggestion is equivalent to our current definition of reconstructability. Also, to further clarify our point, we refer the Reviewer to Supplementary Fig. 1(a) and 1(b), showing that $U(G|X)$ actually scales just like the reconstruction performance of reconstruction algorithms.

Comment 3.4

As I mentioned already, the manuscript is well written and organized. Also, I appreciate the fact that mathematical details are postponed in the appendix and SI. In spite of this, I still find paper excessively long and dense. Somehow, this makes it a little difficult to grasp the main message of the paper, which instead is very intuitive, elegant and important. To increase the future impact of the work and make it accessible to a larger/less specialist readership, I would try to make the paper more concise.

Reply 3.4

We appreciate that the Reviewer finds our manuscript well written and organized, and that our framework is intuitive, elegant and important. We understand the Reviewer's concern about the length and density of the manuscript.

In the last round of reviews, we made significant efforts to add clear interpretations to the measures (Section 2.A) and simple intuitive examples (Section 2.C). As the Reviewer is likely aware, Reviewers 1 and 2 also asked that we move content from the Appendix into the main text (Section II.B and the statement of Theorem 1), and that we extend our analyses to real data (Fig. 7). If any content would have to be removed, we believe it would be any of these. However, removing any of these sections would either reduce the framework clarity or decrease its extent. In the end, we believe that they are necessary to make the manuscript accessible to a larger audience. For these reasons, we chose to leave the manuscript as it is, and hope that the Reviewer understands our decision.

Comment 3.5

The authors may want to cite papers on predictability methods based on maximum entropy sampling: J. Mach. Learn. Res. 9, 235 (2008), Phys. Rev. Lett. 120, 198301 (2018).

Action taken 3.5

We thank the reviewer for pointing out these valuable references, which were added in the Introduction of the manuscript.